# Photocobilins integrate B$_{12}$ and bilin photochemistry for enzyme control

Shaowei Zhang [1,2] ✉, Laura N. Jeffreys [1], Harshwardhan Poddar [1], Yuqi Yu[1], Chuanyang Liu [2], Kaylee Patel[1], Linus O. Johannissen [1], Lingyun Zhu[2], Matthew J. Cliff [1], Cunyu Yan[1], Giorgio Schirò[3], Martin Weik[3], Michiyo Sakuma[1], Colin W. Levy[1], David Leys [1], Derren J. Heyes [1] ✉ & Nigel S. Scrutton [1] ✉

Photoreceptor proteins utilise chromophores to sense light and trigger a biological response. The discovery that adenosylcobalamin (or coenzyme B$_{12}$) can act as a light-sensing chromophore heralded a new field of B$_{12}$-photobiology. Although microbial genome analysis indicates that photoactive B$_{12}$-binding domains form part of more complex protein architectures, regulating a range of molecular–cellular functions in response to light, experimental evidence is lacking. Here we identify and characterise a sub-family of multi-centre photoreceptors, termed photocobilins, that use B$_{12}$ and biliverdin (BV) to sense light across the visible spectrum. Crystal structures reveal close juxtaposition of the B$_{12}$ and BV chromophores, an arrangement that facilitates optical coupling. Light-triggered conversion of the B$_{12}$ affects quaternary structure, in turn leading to light-activation of associated enzyme domains. The apparent widespread nature of photocobilins implies involvement in light regulation of a wider array of biochemical processes, and thus expands the scope for B$_{12}$ photobiology. Their characterisation provides inspiration for the design of broad-spectrum optogenetic tools and next generation bio-photocatalysts.

Photoreceptor proteins are ubiquitous in nature and regulate a large range of biological processes in response to light. They have become essential components for optogenetic applications, where they can be fused to different output domains to provide light control of various cellular functions[1]. The number of known photoreceptor families in biology has recently been expanded[2] by the discovery of a new superfamily of B$_{12}$ photoreceptors that repurpose coenzyme B$_{12}$ or adenosylcobalamin (AdoCbl) for light sensing. Cobalamins are complex cobalt-containing tetrapyrrole molecules, in which various forms differ in the nature of the upper axial ligand (e.g., methyl, cyano, hydroxyl or adenosyl) that is ligated to the central Co atom. For a long time, cobalamin has only been known as a widespread organometallic cofactor to many thermally-driven enzymes that catalyse a wide range of processes essential to living organisms, including humans[3]. The

discovery of CarH, the canonical AdoCbl photoreceptor, revealed B$_{12}$ can support a light-mediated repression of carotenoid biosynthetic genes[4]. In the dark, the AdoCbl-bound CarH tetramer specifically binds to DNA and blocks transcription. On exposure to light, the photolabile Co-C bond of AdoCbl is cleaved, which results in the release of free 4',5'-anhydro-adenosine[4,5]. B$_{12}$ photochemistry triggers structural changes culminating in the formation of a bis-His ligated light state of CarH and the dissociation of the tetramer, together with the concomitant release from DNA and ultimately transcription initiation[4]. CarH has already been used in several green light-dependent biotechnological applications, such as in the formation of light-responsive hydrogels for drug delivery[6–9], light-activated technology devices[10] and the regulation of mammalian gene expression[11]. However, the optogenetic potential of B$_{12}$ photoreceptors would be significantly

[1]Manchester Institute of Biotechnology and Department of Chemistry, The University of Manchester, 131 Princess Street, Manchester M1 7DN, UK. [2]Department of Biology and Chemistry, College of Sciences, National University of Defense Technology, Changsha, China. [3]Univ. Grenoble Alpes, CEA, CNRS, Institut de Biologie Structurale, F-38044 Grenoble, France. ✉e-mail: shaowei.zhang@nudt.edu.cn; derren.heyes@manchester.ac.uk; nigel.scrutton@manchester.ac.uk

enhanced by broadening the wavelength range to include the more penetrative red/far-red region of the spectrum. More complex putative $B_{12}$ photoreceptors have since been identified in genomes, but lack functional characterisation[12,13]. Sequence analysis reveals many CarH-like homologues form considerably more complex protein architectures, with the $B_{12}$-binding domain fused to additional enzyme or chromophore-binding domains[12]. Based on these previous findings, we describe a new subfamily of $B_{12}$-dependent photoreceptors, consisting of a $B_{12}$ binding domain fused to a globin-like (or photoglobin[6]) domain. The hypothesis that these coupled photoglobin and $B_{12}$-binding domains may act as a light-sensing regulatory bundle[13] provides exciting opportunities to develop new optogenetic tools. We now demonstrate that these proteins can simultaneously bind $B_{12}$ and the linear tetrapyrrole biliverdin (BV), and are sensitive to light across the entire visible spectrum (Fig. 1). Hence, we propose these form a new family of photocobilins (**photo**active-**co**balamin-**bilin**, Pcob) that can mediate light-regulation of a range of processes.

## Results and discussion

### Photocobilin proteins exhibit spectral changes upon light activation

We have identified and characterised two types of photocobilins, one from *Saccharothrix syringae* (*Sas*Pcob) that represents a standalone photocobilin photoreceptor and a more complex protein that consists of a photocobilin diguanylate cyclase (DGC) fusion from *Acidimicrobiaceae bacterium* (*Ab*DPcob) that is representative of a photocobilin domain fused to an enzyme domain (Fig. 1 and Supplementary Figs. 1 and 2). Both proteins are able to bind AdoCbl and BV, although there are significant differences in BV binding between the two proteins (Figs. 2a, b, 3). In *Sas*Pcob, the ternary AdoCbl-BV complex is readily formed when both chromophores are provided. AdoCbl binding allosterically affects BV affinity, with the BV dissociation constant decreasing ~10-fold from $K_d$ ~ 7 μM to ~0.7 μM in the presence of AdoCbl (Fig. 3 and Supplementary Fig. 3). In the full-length *Ab*DPcob, full occupancy of AdoCbl is observed in the absence of BV, but the binding of both chromophores simultaneously is diminished compared to the *Sas*Pcob (Fig. 3 and Supplementary Fig. 3). The weaker binding and lower occupancy levels (~20%) of BV in *Ab*DPcob suggest that evolutionary pressures may have reduced the requirement for BV and resulted in a different role for the globin-like domain in these more complex fusion proteins. Despite these differences, both the standalone photocobilin (*Sas*Pcob) and diguanylate cyclase fusion (*Ab*DPcob) show similar light-sensing behaviour (Fig. 2c, d and Supplementary Fig. 4a, b). Illumination of AdoCbl-only bound photocobilin with red light (660 nm LED) has no effect (Fig. 2 and Supplementary Fig. 4), whereas illumination with green light (530 nm LED) elicits spectral changes similar to those observed previously with CarH[14]. These include the formation of new absorbance features at 356 nm and 500 nm, together with the disappearance of the absorbance band at approximately 570 nm. In contrast, the BV-only bound species does not display any obvious spectral changes after illumination with green or red light (Fig. 4a, b). However, the ternary AdoCbl-BV-photocobilin

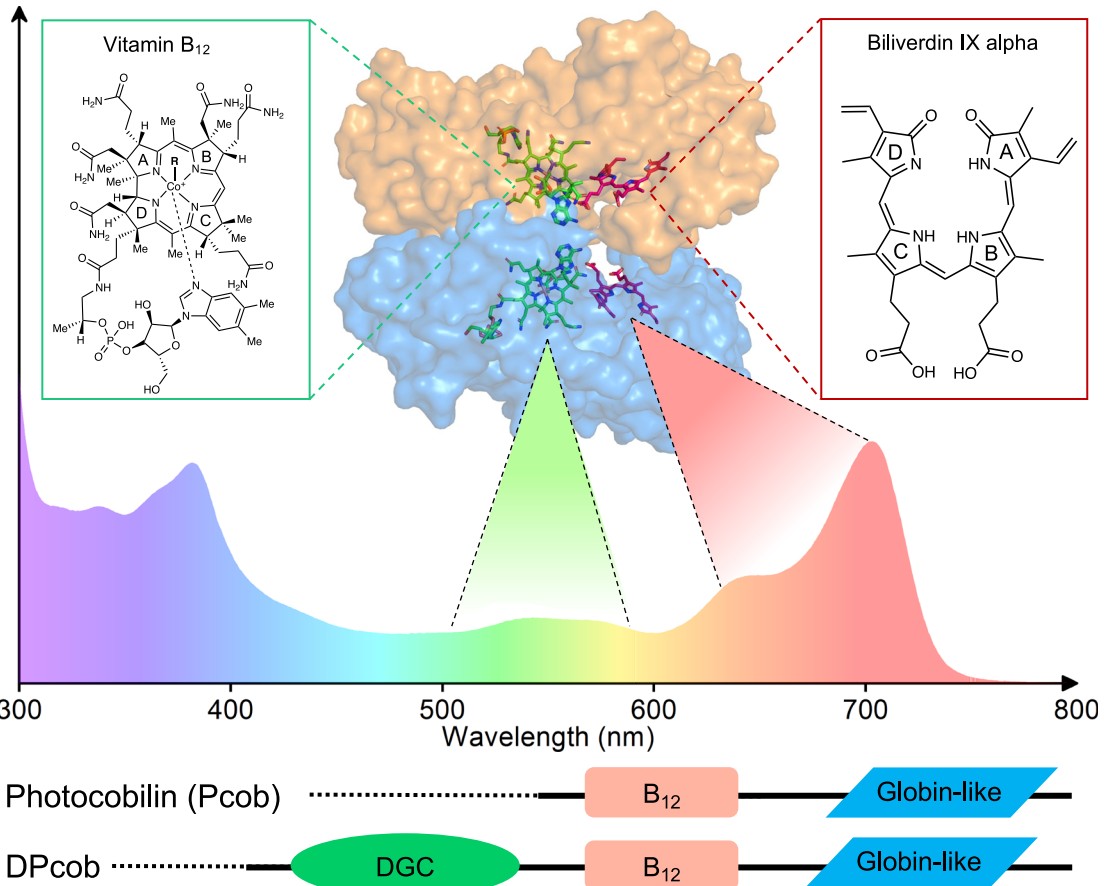

**Fig. 1 | The chromophores and wavelength sensitivity of the new multi-centre photocobilin photoreceptors.** Schematic showing the structures of the adenosylcobalamin (AdoCbl or coenzyme $B_{12}$) and biliverdin (BV) chromophores in photocobilin, together with the wavelength range over which they absorb. The photocobilins are found as both standalone proteins or fused to other domains (a diguanylate cyclase or DGC domain is shown). The nature of the upper R ligand in $B_{12}$ is responsible for the different $B_{12}$ analogues (R=adenosyl, AdoCbl; = CH_3, methylcobalamin; = OH, hydroxycobalamin; =CN, cyanocobalamin). The protein (surface is coloured blue and orange by chains) and absorbance spectrum shown in the figure is from the *Sas*Pcob protein with bound AdoCbl and BV under dark conditions.

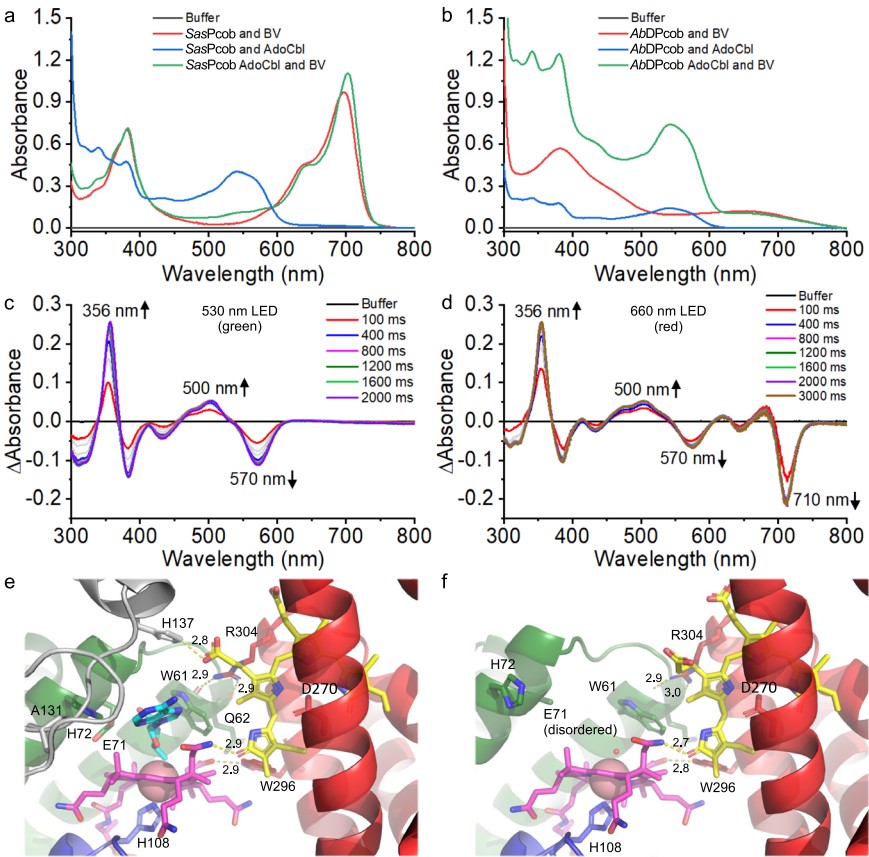

**Fig. 2 | The light response and structure of the photocobilin photoreceptors. a** absorbance spectra of *Sas*Pcob with either BV, AdoCbl or both bound. **b** absorbance spectra of *Ab*DPcob with either BV, AdoCbl or both bound. Difference absorbance spectra of photocobilin photoreceptors after illumination with green light (**c**) or red light (**d**) for *Sas*Pcob. Interaction of BV and AdoCbl molecules in *Sas*Pcob in dark (**e**) and light (**f**) state. The Rossmann fold region is coloured blue, four-helix-bundle region in green and BV binding region in red. The protein main scaffold is shown as cartoon and coloured as above. The opposite monomer is shown as grey cartoon. Main residues involved in the binding are shown as sticks. The AdoCbl and BV molecules are shown as pink and yellow sticks, and the upper ligand of AdoCbl is shown as cyan sticks, respectively. The N and O atoms in the sticks are coloured blue and red, respectively. The cobalt and water molecules as pink and red spheres, respectively. The polar interactions are shown as yellow dash and labelled with distance (angstrom).

complexes respond to both green and red light, with excitation of the BV chromophore leading to spectral changes associated with Co-C bond cleavage in the $B_{12}$ cofactor (Fig. 2 and Supplementary Fig. 4). Hence, the two chromophores are optically coupled and excitation of either can trigger AdoCbl photoconversion, thereby expanding light sensitivity to cover most of the visible spectrum. The relative efficiency of conversion to the light state is higher for the direct AdoCbl excitation route (Supplementary Fig. 5). The ability to respond to red light is lost when AdoCbl is replaced with methylcobalamin, which contains a smaller methyl group as upper axial ligand (Fig. 4c–h). This indicates that the upper adenosyl ligand is required for optical coupling and cleavage of the upper axial C-Co bond. However, the exact mechanism of coupling between the 2 chromophores is currently not clear and will require further characterisation.

## X-ray crystallography provides a structural rationale for photoactivation in photocobilins

To provide a structural rationale for the photocobilin photo-sensing properties, we determined the crystal structures of the dark (PDB 8J2W) and light-adapted states (PDB 8J2X) of *Sas*Pcob, as well as the dark state of the isolated *Ab*DPcob photoreceptor domain (*Ab*Pcob, PDB 8J2Y), at 1.7 Å;, 2.0 Å; and 2.3 Å;, respectively (Figs. 2e, f, 5a–d, Table 1 and Supplementary Fig. 6). Unfortunately, we were unable to obtain crystals of the full length *Ab*DPcob protein or the light state of *Ab*Pcob. Despite the low sequence identity (~34%), the structures of the two photocobilin proteins are similar, with an RMSD value of 1.63 Å for

464 C-alphas after structural alignment. Both the *Sas*Pcob and *Ab*Pcob proteins contain the W-(10)x-EH and E/DxH motifs (Fig. 5e) in the $B_{12}$-binding domain, which is found in other light dependent CarH-like $B_{12}$ photoreceptors[15]. The arrangement and dimerisation of the $B_{12}$ binding domains, which consist of a Rossmann fold and associated four-helix bundle, is similar to the corresponding CarH dimer core module (RMSD = 1.37 for 237 C-alphas), with the dark state *Sas*Pcob and *Ab*Pcob forming head-to-tail dimers[4] (Fig. 5d, e). AdoCbl binding is also reminiscent of CarH, with the dimethylbenzimidazole tail embedded in the Rossmann fold (Figs. 2e, f, 6a), while the conserved His108 (*Sas*Pcob numbering) provides the lower axial ligand to the Co atom of the $B_{12}$ cofactor (i.e., base-off His-on ligation, Fig. 7a–c). The adenosine binding pocket is strikingly similar, with the ribose moiety stacking with the conserved Trp61 residue and forming a network of polar interactions linking across the dimer interface via the conserved Glu71-His72 motif (Figs. 2e, f, 6a). The Pcob specific C-terminal five-helix globin domain does not contribute directly to the dimer interface, and is positioned such that it forms interactions with both of the $B_{12}$-binding domains (i.e., Rossmann fold and four-helix bundle). This arrangement effectively shields much of the corrin ring edge that is normally solvent exposed when bound by the canonical CarH domain.

In *Sas*Pcob, the BV is bound with a *ZZZ* configuration, with Asp270 forming key interactions with the co-planar A, B and C pyrrole rings (Fig. 6a, b). In comparison to other BV-binding photoreceptors, notably the bacteriophytochromes, the photocobilins lack the Cys residue that covalently links to the BV molecule and is essential for light-

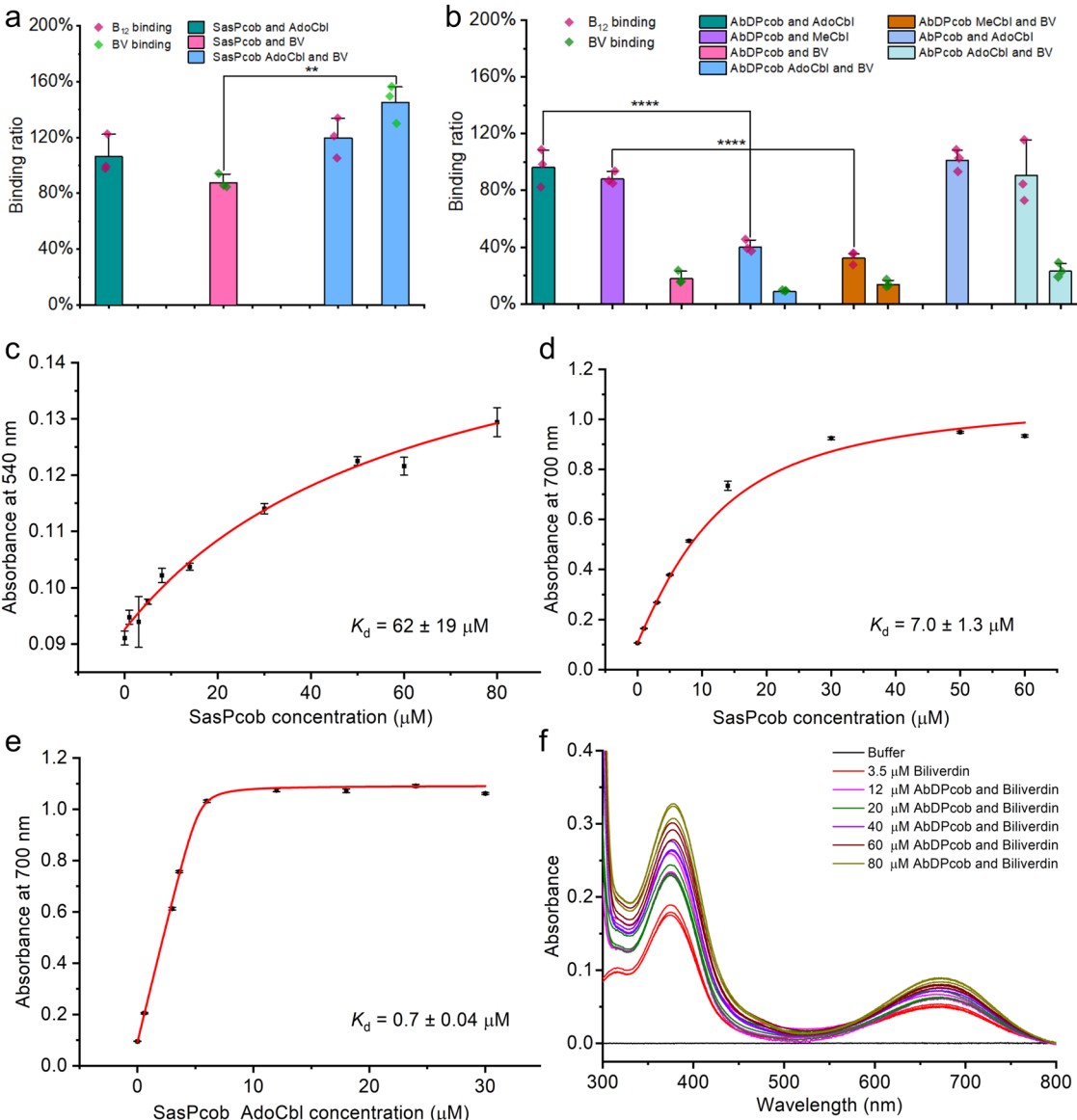

**Fig. 3 | Chromophore binding properties of photocobilins. a** The binding ratio (chromophore:protein) of $B_{12}$ and BV for SasPcob. The binding ratio was calculated based on the chromophore and protein molar concentrations. Ordinary one-way ANOVA F-test with multiple comparisons was performed, F (3, 8) = 11.3, $P = 0.0021$ **$P < 0.01$, $n = 3$ independent experiments. **b** The binding ratio of $B_{12}$ and BV for AbDPcob protein. In (**a**, **b**) values are expressed as mean ± SD. Ordinary one-way ANOVA F-test with multiple comparisons was performed, F (9, 20) = 51.23, $P < 0.0001$, ****$P < 0.0001$, $n = 3$ independent experiments. **c** The absorbance

increase of AdoCbl at 540 nm at increasing concentrations of SasPcob was used to calculate the binding constant ($K_d$) for AdoCbl to the protein. The absorbance increase of BV at 700 nm at increasing concentrations of SasPcob was used to calculate the ($K_d$) for BV in the absence of AdoCbl (**d**) and with AdoCbl pre-bound (**e**). **f** BV titration with AbDPcob. BV was kept at 3.5 µM and AbDPcob ranging from 12 to 80 µM. All measurements were repeated 3 times for data collection. All data are presented as mean values ± SDs, $n = 3$ independent experiments. Source data are provided in Source Data file.

induced *cis-trans* isomerisation[16–18] (Figs. 2e, f and 5f–i). Despite the lack of cysteinyl-linkage, the structure of the SasPcob BV-binding pocket is akin to light-harvesting phycobilisome proteins, with the BV orientation flipped by ~180° (Fig. 5g), projecting the D ring out of the globin core, as opposed to the A ring in other bilin binding proteins[19–21]. The D ring is positioned out of the ABC plane by approximately 40° due to close interactions with Trp296 (Figs. 5g and 6a, b). Although the putative BV binding globin scaffold is retained in AbPcob, the reduced BV affinity is most likely due to replacement of a key BV binding residue, Asp270 in SasPcob, with a His (Fig. 5f–i). The crystal structure of AbPcob revealed that steric hindrance by this His residue interferes with BV binding, unlike in SasPcob. Molecular dynamics simulations and ligand docking (Supplementary Figs. 7–9, Supplementary Table 1 and 2, and Supplementary Data 1) demonstrate that

rearrangement of surrounding residues could allow BV to bind with AbPcob in certain conformations, where hydrophobic interactions with residues in the binding pocket (e.g., Leu252, Trp289, Leu294, Arg297, Val299 and Val303) contribute to retain BV in the correct orientation (Fig. 6c–h). The lack of the main binding interaction with the Asp residue in AbPcob may explain the absence of any obvious spectral shift and increase in absorbance at 700 nm upon BV binding that is observed for SasPcob (Fig. 2). These findings could explain the lower BV occupancy levels for AbPcob whilst retaining the ability to respond to red light (Fig. 3 and Supplementary Fig. 4).

In SasPcob, the close interaction and relative position of the $B_{12}$- and BV-binding domains brings the two chromophores in close proximity, with the edge of the BV C and D rings within van der Waal's distance of the conserved Trp61 and the corrin ring B, respectively.

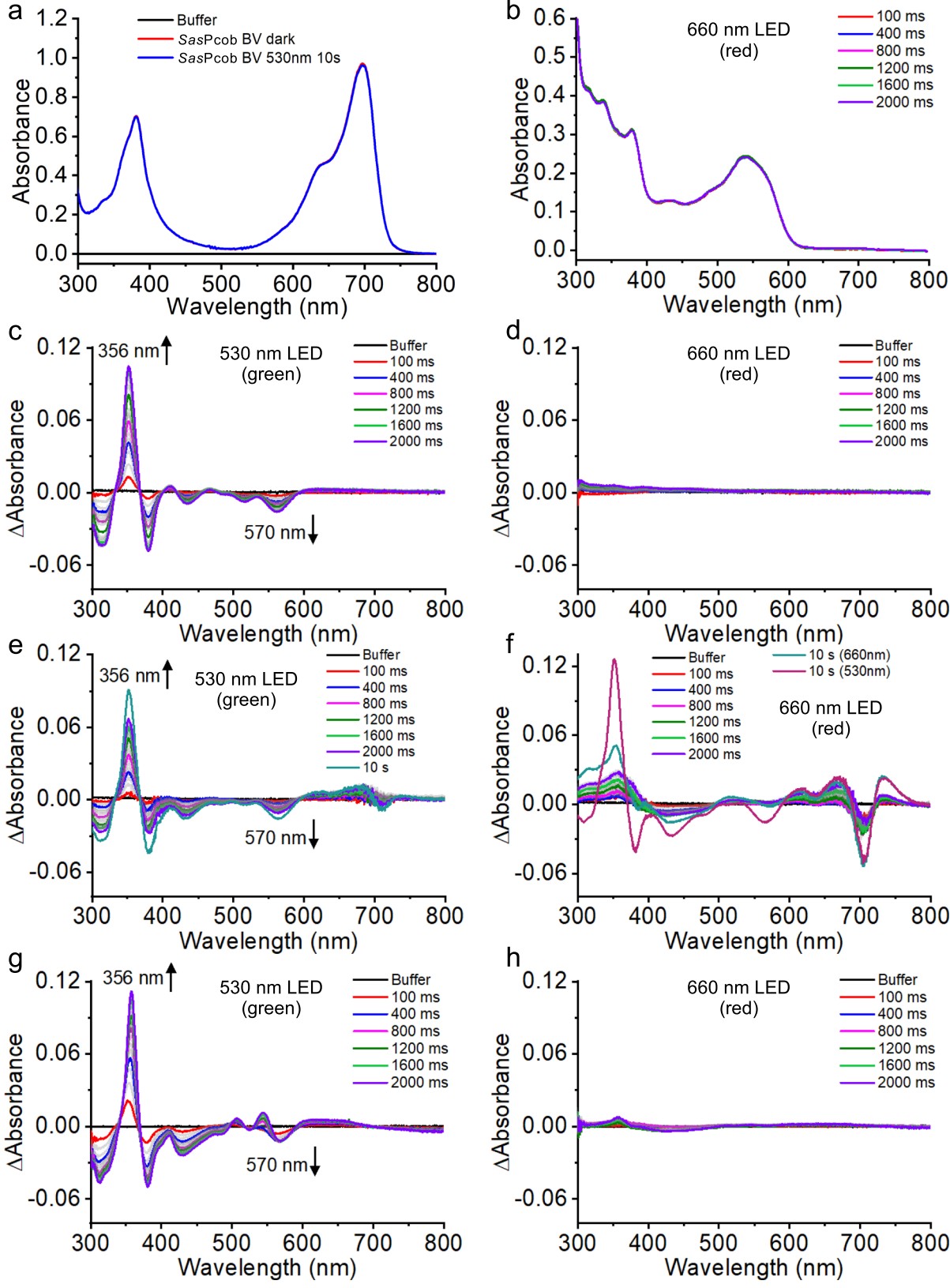

**Fig. 4 | Light response of photocobilin proteins with methycobalamin bound.**
**a** Absorbance spectra of *Sas*Pcob with only BV bound illuminated at 530 nm.
**b** Absorbance spectra of *Sas*Pcob with only AdoCbl bound illuminated at 660 nm.
Difference absorbance spectra of *Sas*Pcob with methycobalamin bound after
illumination with green (**c**) or red (**d**) light. Difference absorbance spectra of

*Sas*Pcob with methylcobalamin and BV bound after illumination with green (**e**) or
red (**f**) light. Difference absorbance spectra of *Ab*DPcob with methylcobalamin and
BV bound after illumination with green (**g**) or red (**h**) light. A dark spectrum was
collected prior to any illumination and used as the blank in each case. The up and
down arrows indicate the increase and decrease in absorbance, respectively.

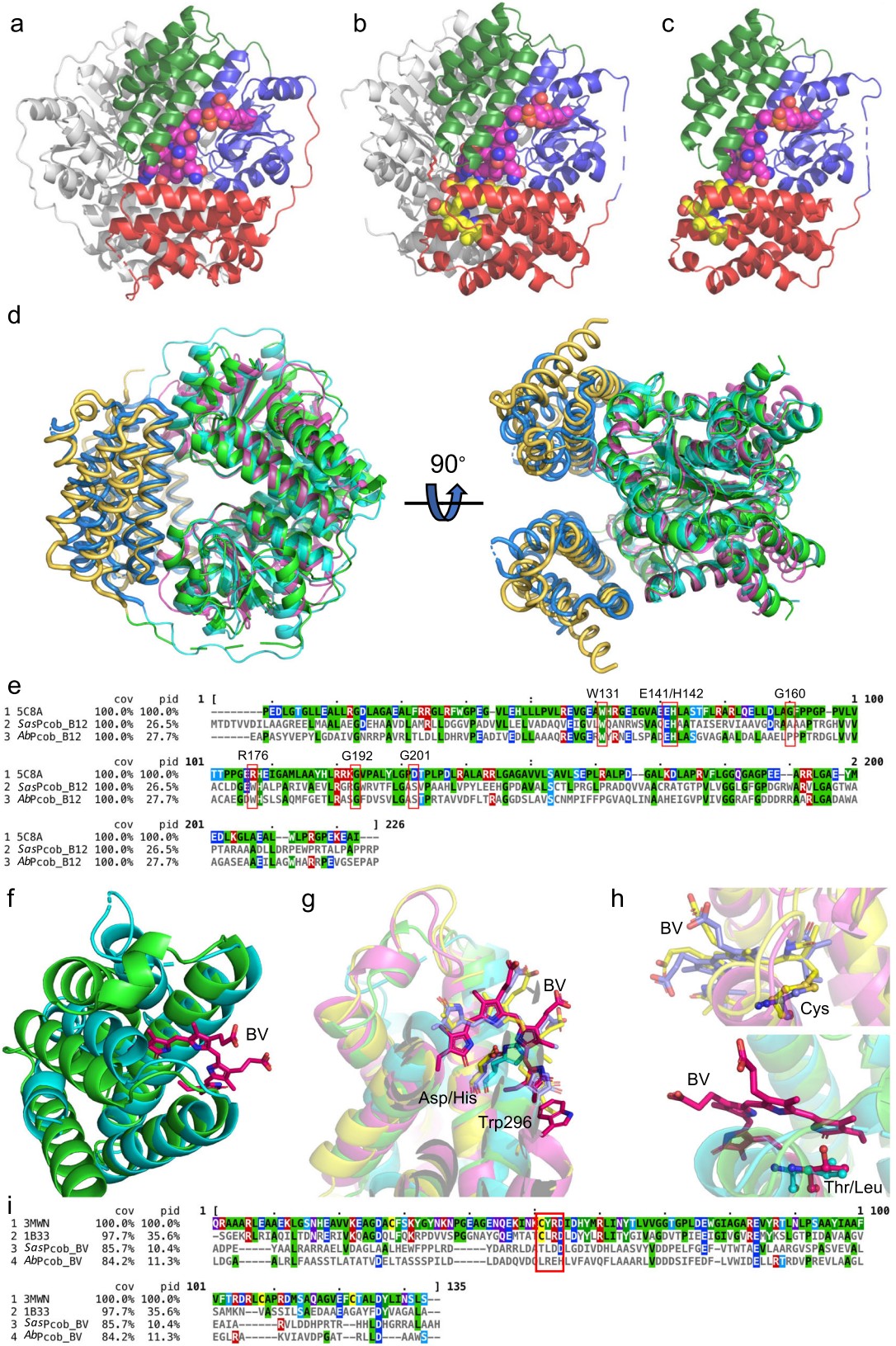

Furthermore, a direct hydrogen bond is formed between one of the corrin ring B amide groups and the BV ring D (Fig. 2e, f). This likely explains the B$_{12}$ allosteric effects on BV affinity observed in solution. A network of direct and water mediated polar interactions surround this central chromophore edge-to-edge contact. These include direct hydrogen bonds between the corrin ring and Trp296 from the globin domain, and a putative salt bridge formed between the propionic acid of ring C of BV and His137 of the opposite monomer (Fig. 2e). The juxtaposition of both chromophores likely underpins the optical coupling observed.

In order to investigate the effects of illumination on the photo-cobilins, we determined the light-exposed *Sas*Pcob crystal structure

**Fig. 5 | Structural comparison of photocobilins with CarH[4] and BV binding protein[20,21].** Overall structure of the dark states of the *Ab*PCob (**a**) and *Sas*Pcob (**b**) protein, and the light state of *Sas*Pcob (**c**). The Rossmann fold region is coloured blue, four-helix-bundle region in green and BV binding region in red. The AdoCbl and BV molecules are shown as pink and yellow spheres. **d** Structural alignment of *Sas*Pcob, *Ab*Pcob and CarH proteins (5C8A). The B$_{12}$ binding region is shown as green (*Sas*Pcob), cyan (*Ab*Pcob), and magenta (CarH) ribbons. The BV binding domain is shown as yellow (*Sas*Pcob) and blue (*Ab*Pcob) ribbons. **e** Sequence alignment of *Sas*Pcob, *Ab*Pcob and CarH proteins. The alignment result is coloured according to sequence identity by MView[56]. The residues in the red box are the key residues at the dimer interface. **f** Structural alignment of BV binding domain of *Sas*Pcob (green) and *Ab*Pcob (cyan). The BV molecule is shown as red sticks. **g** Structural alignment of BV binding domains of *Sas*Pcob (green) and *Ab*Pcob (cyan) with phycobilisome proteins 1B33[20] (yellow) and 3MWN[21] (magenta). BV molecules and key residues involved with BV binding are shown as red (*Sas*Pcob), cyan (*Ab*Pcob), yellow (3MWN), light-blue (1B33) sticks. **h** Comparison of the BV chromophore in the binding pocket. Cys residues bonding with BV (Top, 1B33 and 3WMN), and residues at same position (bottom, *Sas*Pcob and *Ab*Pcob) are shown as sticks. The same colour scheme is used as in (**g**). The BV is shown in red for *Sas*Pcob, yellow for 1B33[11] and blue for 3MWN[12]. **i** Sequence alignment of photocobilins with 3MWN and 1B33. The residues in the red box forms the BV binding motif.

(Figs. 5, 7). Similar to CarH, the *Sas*Pcob light state corresponds to a monomeric species (Supplementary Fig. 1), with the B$_{12}$-binding domain dimer interface disrupted by the light-triggered release of the adenosine group (Fig. 7 and Supplementary Figs. 10–21). However, the *Sas*Pcob light state structure lacks the drastic large-scale domain repositioning that accompanies formation of a bis-His ligated cobalamin in CarH. Instead, more modest changes occur at the adenosine binding pocket in *Sas*Pcob culminating with the formation of a water/hydroxide ligated Co(III)balamin. While the majority of the Rossmann and globin binding domains are relatively unperturbed, the N-terminal four helix bundle adapts to the removal of the adenosine through changes in the position of the Glu71/His72 containing helix. The C-alpha displacement of ~2.4 Å at the Glu71/His72 region is such that, when superposed on the corresponding dark *Sas*Pcob dimer, it would

lead to a severe clash at the dimer interface (Fig. 7c). Therefore, we postulate that rearrangement of the adenosine binding pocket in response to illumination occurs concomitantly with monomer formation.

## Light-induced functional changes in photocobilins

To investigate how the photocobilin light-response can affect functional change, we investigated the activity of the *Ab*DPcob DGC domain. This domain catalyses the formation of cyclic-di-GMP, involved in regulating a number of cellular processes[22–24], from GTP with the release of pyrophosphate. We successfully expressed the full-length *Ab*DPcob protein and measured DGC activity through the detection of both cyclic-di-GMP and pyrophosphate (Fig. 8 and Supplementary Tables 4 and 5). Initially, the production of cyclic-di-GMP, confirmed by LC-MS, was found to increase upon illumination when compared to the *apo* (i.e., no chromophore bound) and AdoCbl-bound dark states of the protein (Fig. 8a). Steady-state kinetics were measured using a pyrophosphate assay for the *apo* and AdoCbl-bound dark/light states of the *Ab*DPcob protein, as well as the truncated DGC domain only (Fig. 8b). Although the Michaelis constant, $K_m$, for GTP is similar for all protein forms, the apparent turnover number, $k_{cat}$, varies significantly and suggests major differences in DGC activity (Fig. 8b). All full-length *Ab*DPcob forms are more active than the isolated DGC domain, indicating that the presence of the photocobilin domain plays a role in maintaining the DGC activity. The binding of AdoCbl to the full-length protein results in a decrease in DGC activity compared to the *apo* protein. However, upon illumination, there is a > 10-fold increase in apparent turnover number, implying that photoconversion of the AdoCbl-bound photocobilin domain to the light state activates the enzymatic activity of the neighbouring DGC domain. Simple replacement of the AdoCbl with the smaller upper ligand found in methylcobalamin, hydroxycobalamin or cyanocobalamin does not increase DGC activity (Fig. 8c). This implies that the presence of the adenosyl group or changes mediated through AdoCbl photochemistry that lead to the removal of the adenosyl group are important in this dark-inhibition and light-activation process. The oligomerisation state of the protein is not affected by GTP binding so this process is light-mediated (Supplementary Fig. 22).

Our attempts to crystallise the full-length *Ab*DPcob protein were unsuccessful, most likely because of the long flexible linker between the photocobilin and DGC domains. To understand the photo-activation process in more detail, we performed small-angle X-ray scattering (SAXS) measurements using the 'dark' and 'light' states of the full-length protein (SASBDB SASDUS3 and SASDUT3 respectively). The results show a significant difference in the scattering signal between the dark and light state (Fig. 9a–c). From the SAXS signals the radius of gyration Rg and the extrapolated intensity at q → 0 I(0) can be extracted. Upon illumination Rg increases from 42.1 Å (dark) to 52.2 Å (light), thus revealing an expansion of the protein, while I(0) remains almost unchanged ($1.03 \times 10^3$ and $1.08 \times 10^3$ for dark and light state, respectively), indicating that the protein remains in its dimeric form. Fourier transform of the SAXS signals produces the signal distribution function p(r), which reveals a transition from a compact single peaked

### Table 1 | Data collection and refinement statistics (molecular replacement) for Pcobs

|  | *Ab*Pcob dark (8J2Y) | *Sas*Pcob dark (8J2W) | *Sas*Pcob light (8J2X) |
|---|---|---|---|
| Data collection |  |  |  |
| Space group | C 2 2 21 | C 1 2 1 | P 32 2 1 |
| Cell dimensions |  |  |  |
| $a, b, c$ (Å) | 108.773 125.836 124.63 | 132.111 97.5389 72.6624 | 109.667 109.667 98.0743 |
| α, β, γ (°) | 90 90 90 | 90 116.716 90 | 90 90 120 |
| Resolution (Å) | 44.31–2.3 (2.382–2.3) | 75.18–1.70 (1.73–170) | 95.16–1.98 (2.051–1.98) |
| $R_{merge}$ | 0.1766 (0.9439) | 0.0765 (0.8162) | 0.07186 (2.404) |
| $I / \sigma I$ | 6.95 (0.94) | 15.0 (1.1) | 20.64 (0.67) |
| Completeness (%) | 99.86 (99.79) | 99.8 (95.9) | 99.92 (99.79) |
| Redundancy | 13.7 (13.3) | 6.9 (5.6) | 20.6 (20.5) |
| Refinement |  |  |  |
| Resolution (Å) | 44.31–2.3 (2.36–2.30) | 75.18–1.699 (1.74–1.70) | 94.97–1.98 (2.03–1.98) |
| No. reflections | 36278 (2610) | 85824 (6228) | 45541 (3314) |
| $R_{work} / R_{free}$ | 0.2094/0.2366 | 0.1829/0.2118 | 0.1832/0.2003 |
| No. atoms |  |  |  |
| Protein | 4902 | 4997 | 2568 |
| Ligand/ion | 231 | 336 | 141 |
| Water | 131 | 406 | 80 |
| *B*-factors |  |  |  |
| Protein | 66.53 | 35.14 | 61.54 |
| Ligand/ion | 43.60 | 26.55 | 50.37 |
| Water | 46.05 | 39.64 | 56.82 |
| R.m.s. deviations |  |  |  |
| Bond lengths (Å) | 0.027 | 0.011 | 0.011 |
| Bond angles (°) | 2.01 | 1.72 | 1.65 |

Values in parentheses are for highest-resolution shell.

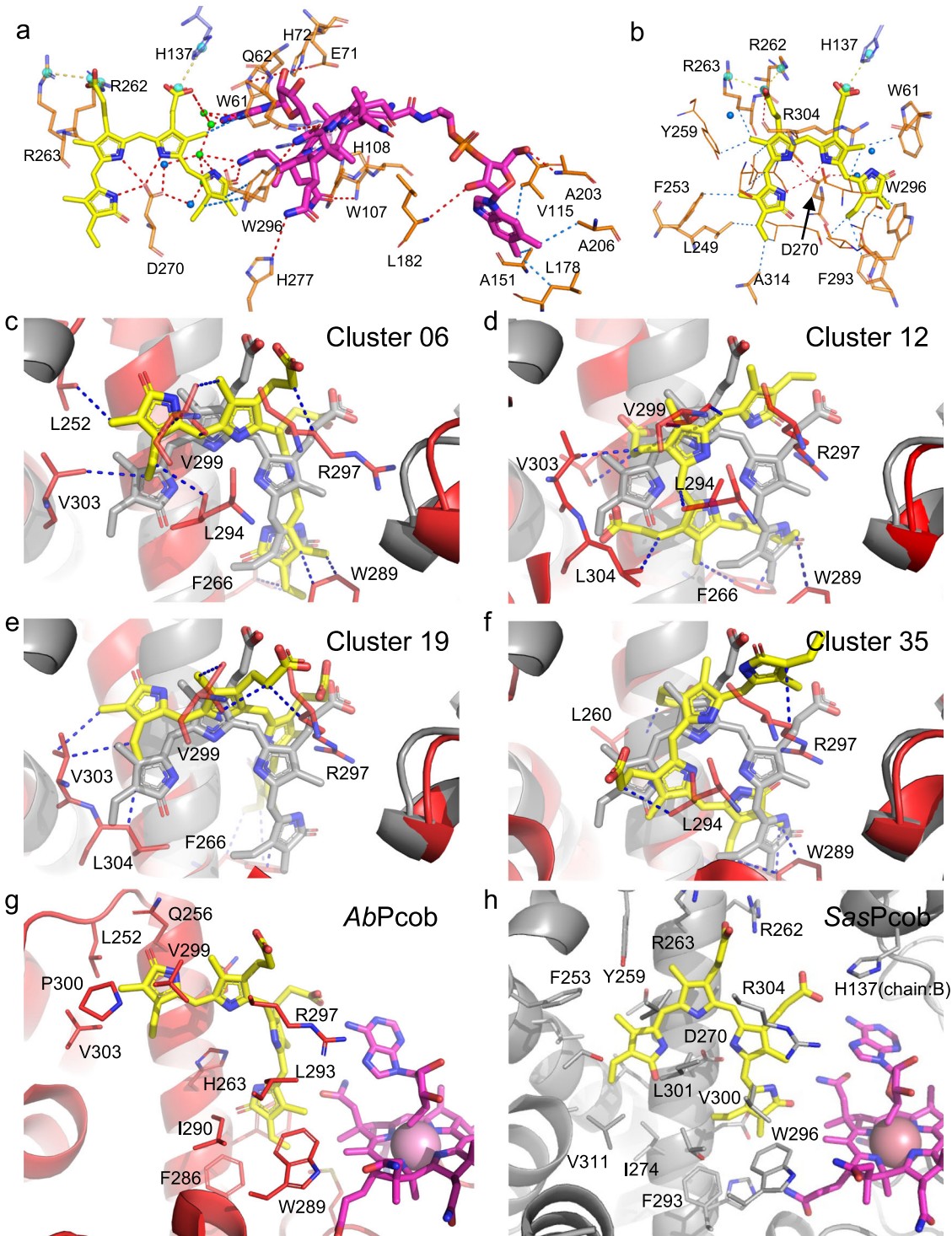

**Fig. 6 | Chromophore binding of *Sas*Pcob and BV docking into *Ab*Pcob.** AdoCbl (**a**) and and BV (**b**) binding in *Sas*Pcob. Protein residues are shown as orange sticks. Residues from the other protein chain are shown as blue sticks. AdoCbl and BV molecules are shown as yelllow and pink sticks, respectively, and water molecules as blue spheres. Structurally-relevant water molecules are shown as green spheres. Salt-bridges and their centres are shown as yellow dashes and cyan spheres, hydrophobic interactions as a blue dash and hydrogen bonds as a red dash. **c**–**f** Clustered *Ab*Pcob protein structures after MD simulations, showing possible binding pose with BV. The grey cartoon and sticks are the representations for

*Sas*Pcob as comparision. BV molecule and its binding region in *Ab*Pcob are coloured as yellow sticks and red cartoon. Residues involved with BV binding are shown as red sticks. The blue dashes indicate hydrophobic interactions between BV and *Ab*Pcob protein residues. Comparison of BV binding in *Ab*Pcob (**g**, modelled) and *Sas*Pcob (**h**, crystal structure). Binding poses in cluster 6 were selected for comparison (see Supplementary Table 1 for clusters summary). BV and B12 molecules are shown as above. Residues around BV with 4 Å are shown as red (*Ab*Pcob) and grey (*Sas*Pcob) sticks.

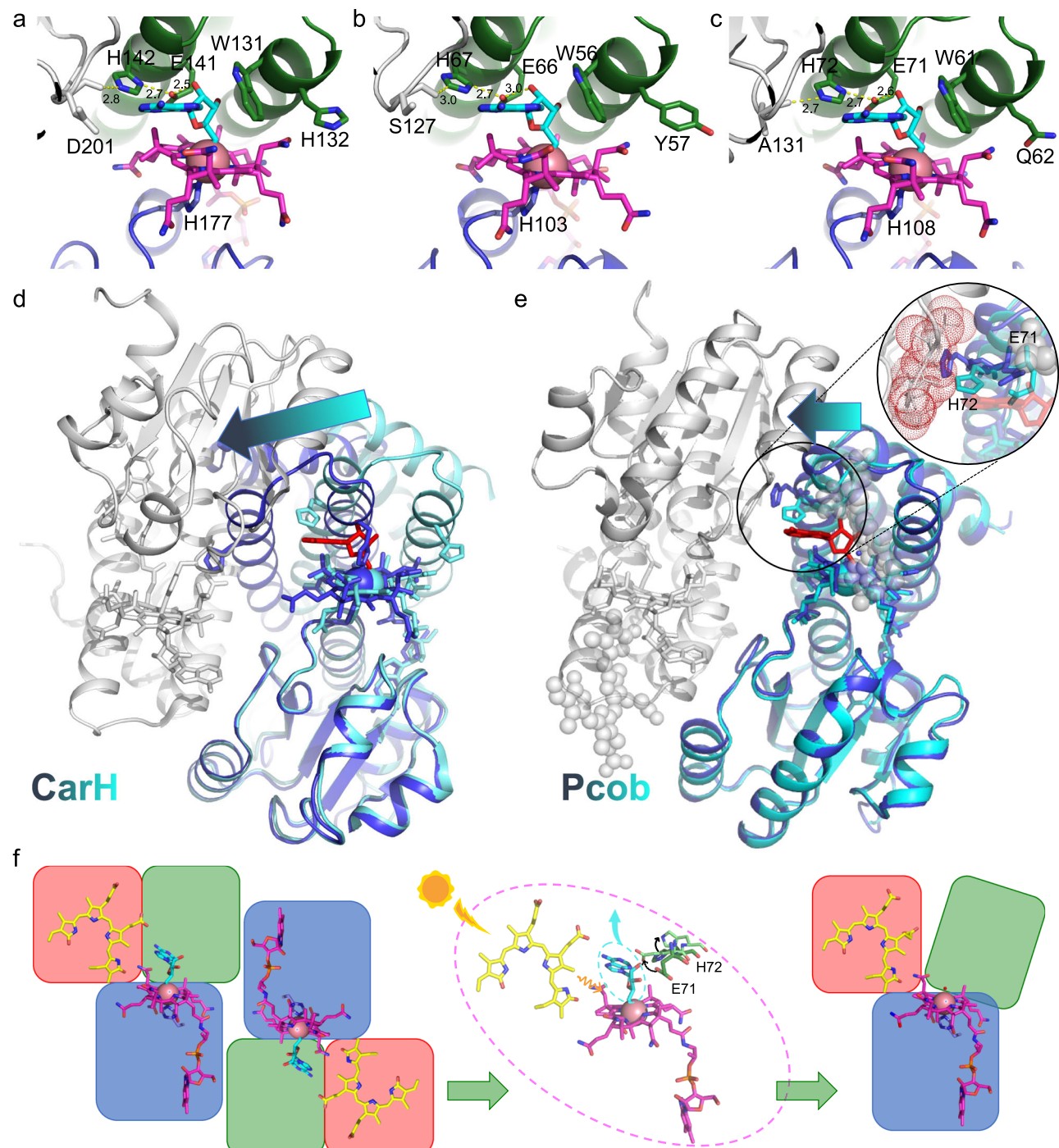

**Fig. 7 | Comparison of AdoCbl binding and light induced changes in photo-cobilin and CarH proteins.** AdoCbl binding in *Tt*CarH (**a**), *Ab*Pcob (**b**) and *Sas*Pcob (**c**) are shown as line and sticks. AdoCbl molecules are shown as magenta and upper ligands as cyan. Residues involved with binding are shown as blue and green sticks. The residues in the opposite monomer are shown as grey. Overall structural changes in *Tt*CarH (**d**) and *Sas*Pcob (**e**). The protein is shown as a cartoon. The main monomer in dark state as blue and light state as cyan. The opposite monomer as grey. The AdoCbl molecules are shown as sticks with dark state as blue and light state as cyan. Upper adenosyl group is shown as red stick. BV molecules in *Sas*PCob are shown as grey spheres. Main residues involved in the change are shown as blue and cyan sticks for dark and light state. The arrow indicates the direction in which the protein moves, while the length of the arrow represents the scale of the movement. The circular perspective provides a more detailed observation of the dynamic motion of residues within dark and light structure. Residues in the opposite monomer are displayed as grey sticks and red dots. **f** Representation of photochemical changes in Pcob protein. The AdoCbl and BV molecules are shown as pink and yellow sticks, adenosyl as cyan sticks. Red, green and blue blocks represent for four helix bundle, Rossmann fold and BV binding regions in the Pcob protein.

distribution for the dark state to a larger multidomain conformation for the light state (Fig. 9b). Low-resolution ab initio modelling based on the p(r) distributions shows that the AlphaFold2[25] prediction for *Ab*DPcob is compatible with the dark conformation in solution, while the light state is characterised by an extended space arrangement largely different with respect to the dark state (Fig. 9c and Supplementary Fig. 23). Molecular dynamics simulations were conducted to compare dynamics of *Ab*DPcob in the light and dark states (Supplementary Figs. 7–9).

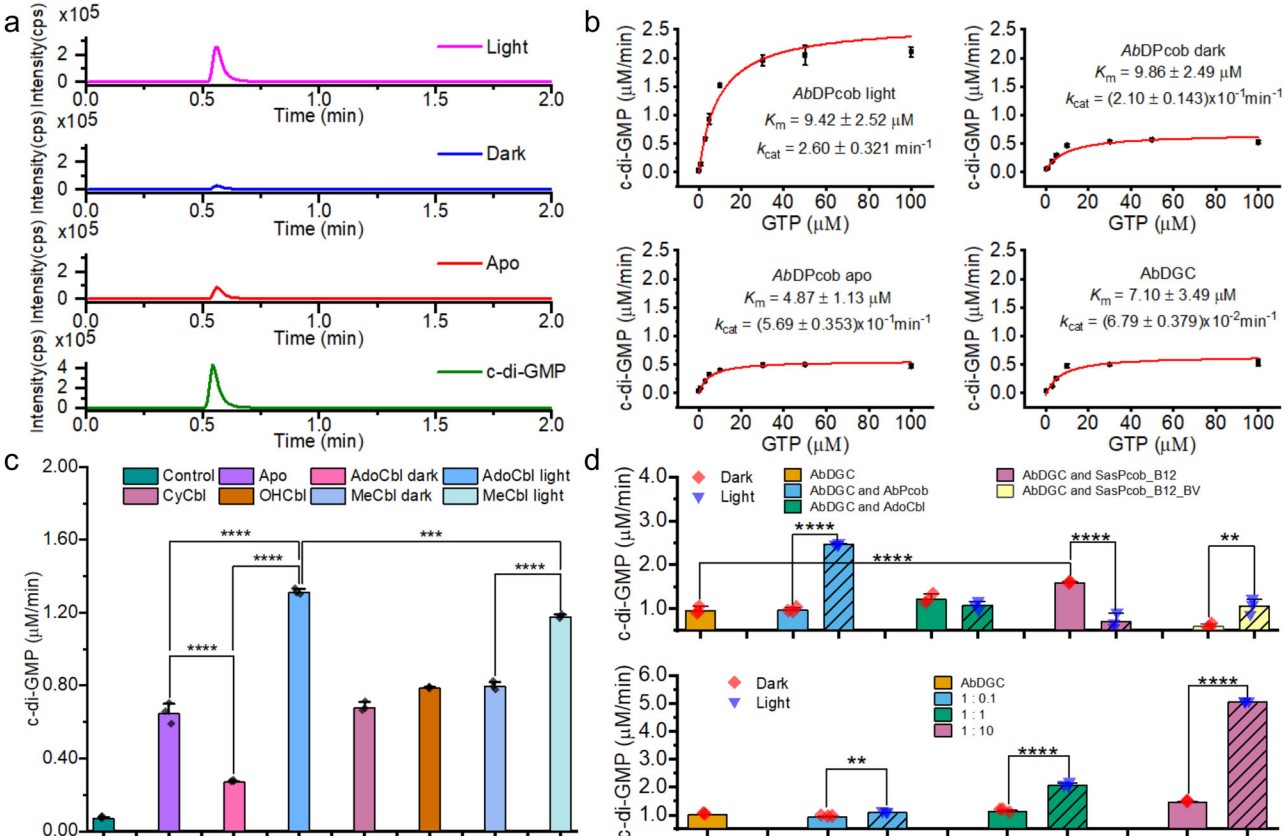

**Fig. 8 | The photoactivation and regulation of enzyme activity by the photocobilins. a** LC-MS chromatograms showing the production of cyclic-di-GMP for the *apo*, dark and light states of the *Ab*DPcob protein (cps, counts per second). Cyclic-di-GMP standard was used for identifying the product. **b** Michaelis-Menten plots showing the rate of cyclic-di-GMP formation for *Ab*DPcob protein under different conditions and the truncated DGC domain only. The rates were normalised for enzyme concentration and fitted to the Michaelis-Menten equation to determine the kinetic parameters. **c** DGC activity of *Ab*DPcob upon addition of different cobalamin cofactors under dark and light condition. Ordinary one-way ANOVA F-test with multiple comparisons was performed, F (7, 16) = 804.7, $P < 0.0001$, ***$P < 0.001$, ****$P < 0.0001$, $n = 3$ independent experiments. **d** The rates of cyclic-di-GMP formation by *Ab*DGC upon addition of the dark or light state of different photocobilin domains (upper panel, including AdoCbl as a control) or different concentrations of the AdoCbl-bound *Ab*Pcob domain (lower panel). The hatch patterns indicate samples for light-adapted states. For upper panel, ordinary one-way ANOVA F-test with multiple comparisons was performed, F (8, 18) = 85.38, $P < 0.0001$, **$P < 0.01$, ****$P < 0.0001$, $n = 3$ independent experiments. For lower panel, ordinary one-way ANOVA F-test with multiple comparisons was performed, F (6, 14) = 4024, $P < 0.0001$, **$P < 0.01$, ****$P < 0.0001$, $n = 3$ independent experiments. All data are presented as mean values ± SDs, $n = 3$ independent experiments. Source data are provided in Source Data file.

*Ab*DPcob showed larger conformational change in the light state compared to that in the dark state indicated by structural deviation (RMSD) from the same starting structure (Supplementary Fig. 8). Specifically, the Pcob domain stays more stable than the DGC domain in both light and dark state. The DGC region was observed to show higher flexibility in the light state particularly at the interface with the Pcob region (Supplementary Fig. 7). Further analysis revealed an obvious opposite movement of the Pcob and DGC domains in the light state but not in the dark state (Fig. 9d, Supplementary Fig. 23 and Supplementary Movie 1 and 2). This indicates that photoactivation of *Ab*DPcob is modulated by local protein conformation changes.

Although a clear photoactivation role has been established for the photocobilin domain fused to the DGC enzyme domain, the function of the standalone photocobilin photoreceptors is less clear. To study this aspect, we measured the activity of the isolated *Ab*DGC domain upon mixing with an equal molar ratio of either of the two photocobilin proteins, isolated AdoCbl-bound forms of *Ab*Pcob or *Sas*Pcob (with and without bound BV) (Fig. 8d). Similar to the previous findings, the light-adapted states of the $B_{12}$-bound *Ab*Pcob and the *holo Sas*Pcob (i.e., both AdoCbl and BV bound) activate *Ab*DGC catalysis. The increase in enzyme activity is dependent on the concentration of the photocobilin protein (Fig. 8d), suggesting a potential role for protein-protein interactions in the activation process. Surprisingly, in the

absence of BV, the AdoCbl-bound *Sas*Pcob dark state activates *Ab*DGC activity, whereas the light state appears to inhibit activity (Fig. 8d). Further evidence that the two proteins interact is provided by size-exclusion chromatography multi-angle light scattering measurements, which show dissociation of the *Ab*DGC dimer into monomers upon addition of the photocobilin domain (*Ab*Pcob). (Supplementary Fig. 2). Higher concentrations of the photocobilin domain give rise to the appearance of a higher molecular weight species, which is likely to represent a heterodimer between the photocobilin and the *Ab*DGC protein. Taken together, these results suggest that the standalone photocobilin photoreceptors can also regulate protein function through protein:protein interactions (PPIs), a mechanism that could potentially influence a wide range of metabolic pathways.

### A wider role for photoglobins in cellular processes
Previous bioinformatics approaches have shown that photoglobin domains can be fused to several other types of functional domain and have proposed that coupled photoglobin and $B_{12}$-binding domains (i.e., photocobilins) may be involved in the regulation of different enzymes and transcription factors[13]. We have now shown that the photocobilins have the ability to regulate enzyme activity through PPIs. As such, we have investigated potential PPI networks to explore possible biological functions in cell metabolism (Supplementary

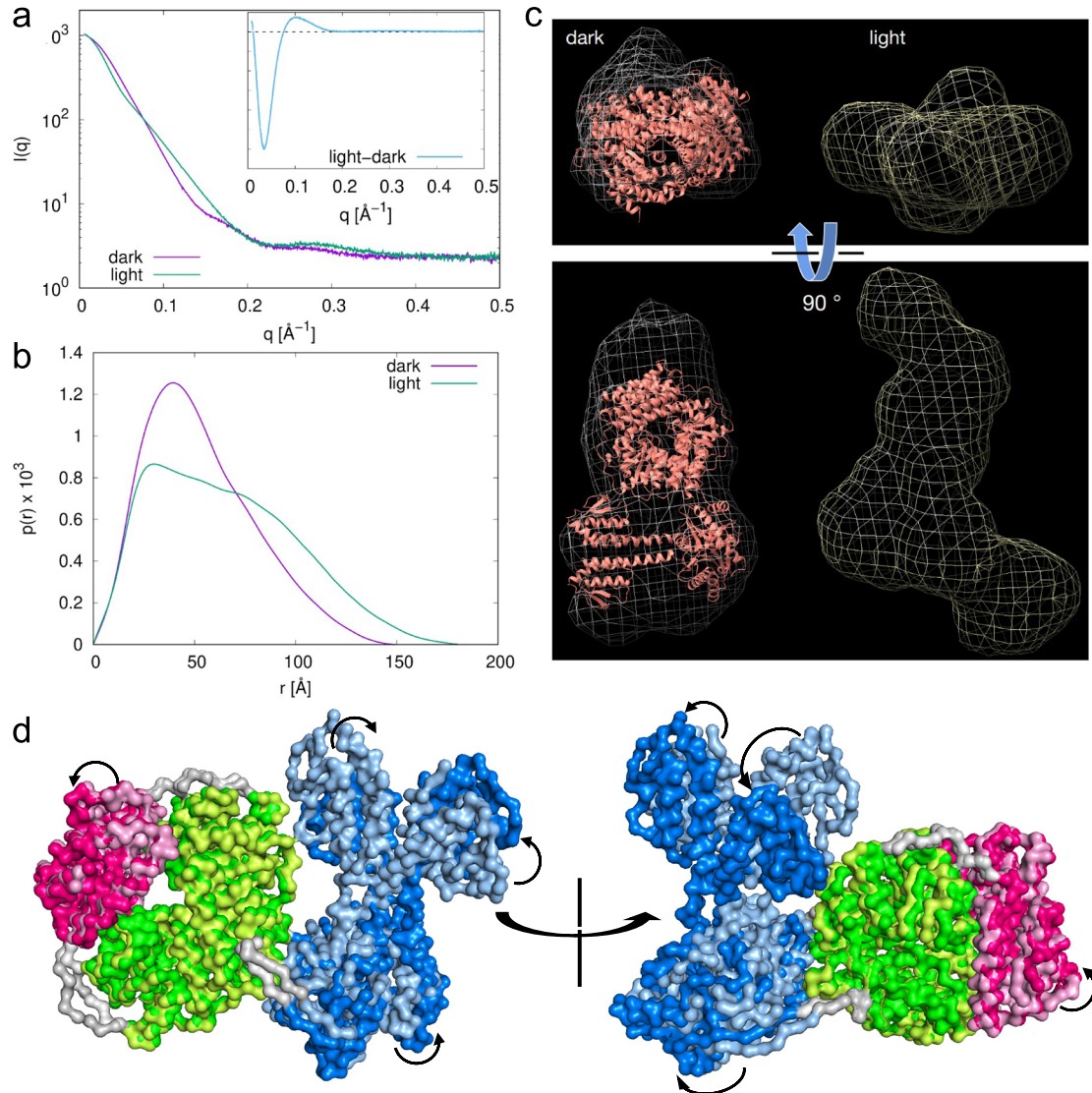

**Fig. 9 | Light induced structural changes in *Ab*DPcob. a** SAXS signals of *Ab*DPcob in the dark state and after illumination at 530 nm for 10 min. The inset shows the difference between the two signals. **b** The distance distribution p(r) reveals an expansion of the protein upon illumination. **c** Low-resolution ab-initio modelling based on the SAXS data provides the envelope of the protein electron density distribution for both dark and light state. Results in (**c**) show that the AlphaFold2 prediction (orange structure) is compatible with the dark structure in solution, while the light state is characterised by an extended conformation. **d** Superposition of *Ab*Pcob in dark and light state form in molecular dynamics simulation results of *Ab*DPcob. AlphaFold2 predicted dimer was used as the starting model. The final dark and light state structures are aligned in Pymol. Protein surface is shown according to domain arrangement. In dark state, $B_{12}$ domain is shown as lemon-green, BV as light-pink and DGC as light-blue. In light state, $B_{12}$ domain is shown as green, BV as hot-pink and DGC as marine-blue. The linker regions in both dark and light state are shown as grey. The arrows in the figure indicate the protein movement upon conversion from the dark to the light state. The SAXS data collection parameters are shown in Supplementary Table 3.

Fig. 24). In *Saccharothrix* species, *Sas*Pcob potentially interacts with STAS (Sulphate Transporter and AntiSigma factor antagonist) domain, Stage II sporulation protein E (SpoIIE) and Sigma factor PP2C-like phosphatases (Supplementary Figs. 24 and 25a), suggesting a putative role in regulation of the cell stress response. Other photocobilins were frequently involved in one carbon pool by folate, amino acid metabolism and purine biosynthesis, suggesting potentially broad participation in cell metabolism (Supplementary Figs. 24 and 25a). Sequence similarity networks shows there are a wide range of photocobilin photoreceptor proteins with a high similarity of overall sequence, structure, motifs and domain architecture, either fused to other enzyme domains or as standalone proteins (Supplementary Figs. 26–30). A comparison of our *Sas*Pcob structure with other photocobilin proteins (predicted structures obtained from the recently published Alphafold2 database[26]), shows they can be divided into three

main groups depending on the location of the AdoCbl-binding domain at either the N- or C-terminus, and whether they contain the additional output domain in the fused proteins (Supplementary Figs. 25 and 31). The majority of the predicted structures align well with *Sas*Pcob (average RMSD = 1.27 Å). Some of these putative photocobilin proteins have variable lengths of the linker region between the AdoCbl- and BV-binding domains, which may affect the arrangement of the overall scaffold, while a small number of the fusion proteins lack a complete photocobilin domain. As the Alphafold2 structural models only show the monomeric form of the protein it is likely that these structures may differ in the higher order oligomeric form of the protein.

## Conclusions

In conclusion, we have identified and characterised a new sub-family of AdoCbl-dependent photoreceptors that uniquely use two different

chromophores to expand the wavelength range of $B_{12}$ for sensing and responding to light. The close proximity of the $B_{12}$ and BV cofactors in the photocobilins allows interaction and 'cross-talk' between them, meaning excitation of BV triggers $B_{12}$ photochemistry. This leads to structural change in the photocobilin protein that can promote enzyme activity either through direct fusion to the enzyme domain or by protein-protein interactions. The photocobilins are likely to have multiple functions in regulating cell metabolism and provides further light-sensing roles for $B_{12}$ in biology. The unique coupling of $B_{12}$ and BV in these novel photoreceptor proteins now opens a window to their exploitation as new optogenetic components.

## Methods

### Protein expression and purification
The proteins were expressed and purified as previously described with slightly modifications[27]. The potential photoreceptor DNA sequences were selected, codon optimised and synthesised by GeneArt® (ThermoFisher) company. Synthesised genes were sub-cloned into a pET21a (Novagen) vector with a C-terminal 6·His tag. The recombinant plasmid was transformed into *E. coli* BL21(DE3) for protein expression. Different IPTG concentrations and auto-induction LB medium were used to optimise soluble expression. The large-scale protein expression was carried out with auto-induction LB media (FormediumTM, glucose/lactose ratio 1:4) containing 50 µg/mL ampicillin. After 24 h incubation at 25 °C, cells were harvested by 10 min centrifugation at 6000 g, 4 °C. Harvested cells were resuspended in 20 mM HEPES pH 7.0, 500 mM NaCl, 25 mM imidazole (lysis buffer) supplemented with protease inhibitor cocktail and then lysed by a cell disruptor at 25 kpsi (Constant Systems). The cell lysate was centrifuged at 20,000 rpm (48,000 × g) for 1 h at 4 °C to remove cell debris. The supernatant was collected and loaded onto a HisTrap column (Cytiva). After washing with cell lysis buffer, bound protein was eluted with 20 mM HEPES pH 7.0, 500 mM NaCl, and 250 mM imidazole. The peak fractions were collected and incubated with ligands (either AdoCbl, MeCbl or BV) for at least 2 h at 4 °C under dark conditions. The sample was then loaded on to a size exclusion column (HiLoad 16/600 Superdex 200) to remove free ligands and further purification. Absorbance spectra were used to confirm ligand binding and protein fractions with ligand bound were collected for further experiments. The CarH protein was expressed and purified for photoproduct determination (Supplementary Figs. 10−21) as described[27].

### Crystallisation, data collection, and structure determination
Purified proteins were exchanged into 20 mM HEPES pH 8.0, 150 mM NaCl and concentrated to 20 mg/ml. Crystallisation was performed in the dark using the sitting drop vapour diffusion technique (200 nL crystallisation reagent mixed with 200 nL protein). The preparation of the dark sample was conducted in the dark with dimmed red light. In order to achieve full conversion to the final light-adapted state, the protein samples were exposed to a 530 nm LED light for a duration of 5 min. Absorbance spectral analysis was performed to ensure complete and effective illumination. The *Sas*Pcob dark crystals were obtained from LMB crystallisation screen B7 with 38% v/v 1,4-dioxane. The *Sas*Pcob light crystals were obtained from 0.2 M Potassium citrate tribasic monohydrate with 20% w/v PEG 3350. The *Ab*Pcob crystals were obtained from 8% w/v PEG 20000, 8% v/v PEG 550 MME, 0.1 M sodium acetate, pH 5.5, 0.2 M potassium thiocyanate. Crystals were cryo- protected by the addition of 20% PEG 200 to the reservoir solution and flash frozen in liquid nitrogen. Individual datasets were collected from single crystals at beamlines i03 (Diamond Light Source). All data were indexed, scaled and subsequently integrated with Xia2 Dials. Structure determination was initially performed by molecular replacement in Phaser[28] using a search model generated by Alphafold2[25]. A combination of automated and manual rebuilding and refinement in Refmac[29] and COOT[30] were used to produce the refined

models. Chromophore ligands were obtained from Refmac monomer library by 3 letter code and refined using COOT[30] and Refmac[29]. Validation with both Molprobity[31] and PDB_REDO[32] were integrated into the iterative rebuild process. Complete data collection and refinement statistics are available in Table 1. The atomic coordinates and experimental data have been deposited in the Protein Data Bank (www.pdb.org). All figures were made using open-source PyMOL 2.5 software.

### LED illumination and absorbance spectroscopy
Absorbance spectra were collected using a Cary 60 spectrophotometer (Agilent Technologies). All measurements were carried out under dimmed red light. The dark spectrum was collected prior to any illumination. A TDS3032C 300 MHz Digital Phosphor Oscilloscope (Tektronix) and TGP110 10 MHz Pulse Generator with Delay (Thurlby Thandar Instruments) were used to generate a 100 ms LED (Thorlabs Inc.) pulse at either 530 nm (green light) or 660 nm (red light). After each LED pulse, the spectrum was collected until there were no further significant changes. The difference spectra were obtained by subtracting the dark spectrum from the illuminated spectrum. Data were normalised and plotted using Origin 9.0 software (OriginLab, Northampton, MA).

### $B_{12}$ and BV binding measurements
The binding ratio of $B_{12}$ and BV ligands was calculated based on their extinction coefficient. AdoCbl, 8.0 mM$^{-1}$•cm$^{-1}$ at 540 nm[33]. MeCbl, 7.7 mM$^{-1}$•cm$^{-1}$ at 540 nm[34]. BV, 90 mM$^{-1}$•cm$^{-1}$ at 700 nm[35]. The protein concentration was determined by utilising the ProtParam tool[36] to calculate its theoretical extinction coefficient derived from its protein sequence. *Sas*Pcob, 55.46 mM$^{-1}$•cm$^{-1}$ at 280 nm. *Ab*DPcob, 63.91 mM$^{-1}$•cm$^{-1}$ at 280 nm. The binding constant was measured by monitoring the spectral changes at increasing protein concentrations. The absorbance change at 540 nm was plotted for $B_{12}$ binding and the absorbance change at 700 nm was plotted for BV binding. With tight binding ligand, the binding constant ($K_d$) is much lower than ligand concentration used in the titration assay, tight binding equation (Eq. 1) should be used to get a more accurate $K_d$ value. BV binding data were fitted to the tight binding equation (Eq. 1) and AdoCbl binding data to apparent binding equation (Eq. 2) to obtain the $K_d$ value using Origin 9.0 software (OriginLab, Northampton, MA).

$$[EL] = \frac{([E]+[L]+K_d) - \sqrt{([E]+[L]+K_d)^2 - 4[E][L]}}{2[L]} + c \qquad (1)$$

$$[EL] = \frac{[E] \times [L]}{K_d + [E]} + c \qquad (2)$$

where [EL] is the concentration of the enzyme and ligand complex formed, [E] is the enzyme concentration and [L] is the concentration of the ligand, c is the constant.

### Diguanylate cyclase activity assay
The diguanylate cyclase activity was determined by using both liquid chromatography−mass spectrometry (LC-MS) and a pyrophosphate assay. For LC-MS, the reaction was prepared with 10 µM enzyme, 200 µM GTP in 20 mM HEPES buffer, pH 6.8, 150 mM NaCl, 10 mM MgCl$_2$. The reaction was carried out at 30 °C for 5 min, then deactivated at 95 °C for 5 min. The dark state reaction was carried out in a black tube and the light state reaction in a transparent tube after illumination with white light. All samples were then cooled down on ice and centrifuged for 15 min at 15,000 rpm. Samples were diluted 50 times with HPLC grade water and transferred to LC-MS glass vials for quantification on a Waters ACQUITY Xevo TQ-S UPLC−MS/MS (Waters Corporation) under negative mode (ESI-). The analysis was carried out using an Acquity UPLC HSS T3 column (1.7 µm, 50 × 2.1 mm) with an

optimised gradient programme. The mobile phases were as follows; A: 10 mM ammonium acetate in water contains 0.1% acetate acid (v/v); B: methanol; flow rate: 0.6 mL/min; column temperature: 30 °C. The gradient starts from 97% A and hold for 0.5 min then decreased to 2% in 0.3 min, hold the 98% B for 0.4 min then returned to 97% A in 0.1 min and equilibrated for 0.7 min for waiting another injection. The detection of cyclic-di-GMP was optimised by employing the multiple reaction monitoring transition of 690.9 > 152.0 to quantify the yield of the reaction. The mass spectrometer parameters were as follows: source desolvation temperature of 600 °C and source ion block temperature of 150 °C were used; Nitrogen (≥95%) desolvation gas was set at 1000 L/h, and nebuliser gas at 7.00 Bar; Argon (zero grade) collision gas flow was set at 0.15 mL/min. The system was trained with a series of standard solutions of cyclic-di-GMP and the limit of detection limit was determined. The quantification standard curve was obtained ranging from 6 nM to 20 μM.

The EnzCheck pyrophosphate assay kit (ThermoFisher) was used to follow the kinetics of diguanylate cyclase activity. The reaction was carried out with 1–10 μM enzyme, 0–100 μM GTP in 20 mM HEPES buffer, pH 6.8, 150 mM NaCl, 10 mM MgCl$_2$. The reaction mixture was incubated in the dark or by illumination with a 530 nm LED for 3 min at 30 °C, prior to the addition of GTP to initiate the reaction. Pyrophosphate standard (0–60 μM Na$_4$P$_2$O$_7$) in the same buffer condition was used to generate the standard curve for accurately quantifying pyrophosphate concentration. The pyrophosphate produced was measured and quantified and then converted to cyclic-di-GMP production rate (two molecules of pyrophosphate equals to one molecule of cyclic-di-GMP). All data were collected in triplicate and plotted against GTP concentration. The $K_m$ was obtained by fitting data into the Michaelis-Menten equation (Eq. 3).

$$v = \frac{V_{max} \times [S]}{K_m + [S]} \tag{3}$$

## SAXS data collection and analysis
Static X-ray scattering data were collected at 20 °C at the BM29 BIO-SAXS beamline of the ESRF Synchrotron (Grenoble, France). X-ray solution scattering signals in the SAXS region (q = 0.0025−0.5 Å$^{-1}$) were collected on AbDPcob at 5 mg/mL concentration using a monochromatic X-ray beam (double multilayer monochromator) centred at 12.5 keV and a Pilatus3 2 M (Dectris) detector. Two datasets were collected, before and after irradiation of the AbDPcob solution with a LED light source at 530 nm for 10 min. For each dataset, 10 X-ray scattering signals (each registered with 1 s of X-ray exposure) were collected with 10 analogous signals of the buffer before and after each protein measurement. Each two-dimensional pattern was converted to a one-dimensional scattering profile by azimuthal integration. Corresponding scattering profiles were averaged and protein signals were obtained by subtraction of the buffer signal. The distance distribution functions p(r) were computed using GNOM[37], and the Rg and I(0) parameters were determined from the reduced data using routines from the ATSAS suite[38]. Low-resolution molecular envelopes were computed using DAMMIF[37].

## Molecular dynamics simulations of AbDPcob and AbPcob in light and dark state
The full-length AbDPcob and AbPcob was modelled using AlphaFold2[25] predicted and solved crystal structure respectively. The AdoCbl and cobalt was placed in AbDPcob by taking the coordinate in the crystal structure of AbPcob after structure alignment to Pcob domain. OHCbl was created by a simple mutation of the upper ligand in PyMol 2.5. We simulate the dynamics of AbDPcob and AbPcob in light and dark state by including different upper ligand into the protein: AdoCbl for dark state and OHCbl for light state. The input files for the calculation of parameters for AdoCbl and OHCbl were generated by MCPB.py protocol[39] in AmberTools 22[40,41] package. The starting structure of AdoCbl from the crystal structure was used as the starting geometry and the starting structure of OHCbl was created by replacing the adenosyl group with OH group in AdoCbl. Both ligands were first optimised by B3LYP/6-31 G*[42] in Gaussian 09[43] and the optimised structure was used for calculating point charges and force constants. The protonation state of protein residues were calculated by PROPKA[44] in the presence of AdoCbl/OHCbl. AMBER99SB force field[45] was applied to model protein and ions. The complex structure was solvated by TIP3P modelled water molecules[46] in cubic box, keeping at least 14 Å between any atoms of protein and the boundary of the box. Counterions (19 Na$^+$ ions for truncated AbPcob and 58 Na$^+$ ions for full-length AbDPcob) were added to neutralise the system. MD simulations were carried out by Gromacs version 5.0[47]. We use the same protocol for the simulation for the protein in light and dark state.

The system was first minimised to remove possible clashes after adding hydrogens to the protein, followed by a heating step to 300 K under NVT ensemble. The initial velocities were generated to maintain a Maxwell−Boltzmann distribution at the desired temperature, 300 K. The stepwise equilibration steps are conducted in order to make sure the structure gradually relaxed in the solvent environment: firstly, protein-complex were restrained to minimise/equilibrate (100 ps) water and ions; secondly, heavy atoms of protein and AdoCbl/OHCbl-Cobalt were restrained to minimise/equilibrate (100 ps) protein hydrogens; next main chain of protein and heavy atoms of AdoCbl/OHCbl and Cobalt were restrained to minimise/equilibrate (2 ns) protein side chains; then, protein Cα atoms and heavy atoms of AdoCbl/OHCbl and Cobalt were restrained to minimise/equilibrate (2 ns) main chain of protein; in last step, all restraints were remove to equilibrate for 2 ns. Harmonic restraint was applied before full relaxation of structures with a force constant of 1000 kJ/mol/nm². Production was run at 300 K by NPT ensemble for 1500 ns based on the equilibrated structure (500 ns for AbPcob). V-rescale thermostat algorithm[48] was used to control the temperature with a response time 1.0 ps. The pressure was kept at 1.0 bar using the Parrinello-Rahman pressure coupling scheme[49] with a response time 1.0 ps. Hydrogen atoms were constrained during the production by LINCS[50]. The cut-off was set to 12 Å for electrostatic interactions calculation. Additional sampling was carried out on AbDPcob using three parallel simulations as follows: 50 × 1 ns cycles of simulated annealing (700 ps at 300 K, 100 ps heating to 400 ps, 100 ps at 400 K, 100 cooling to 300 K) followed by an additional 350 ns of MD for each run, which was then used for analysis. The increased sampling is clear from the Cα RMSDs for these simulations (more details see supplementary information).

## Ensemble docking of BV to AbPcob
The AbPcob conformations in dark state after MD simulation was used for BV docking analysis. AbPcob dark conformations were clustered based on RSMD values of BV binding region. Gromacs gromos[51] method was used for clustering simulated structure at 1.5 Å cut-off. 38 clusters were obtained with RMSD cut-off of 1.5 Å for two structures to be neighbour. Autodockvina[52] was used to dock BV into clustered AbPcob structure with a grid box size of 18 Å × 18 Å × 18 Å. All docked binding poses were ranked by binding affinity. Structures with a value below −7.0 kcal/mol were selected for further docking analysis. To get more detailed binding possibilities, the search exhaustiveness was set to 72 to rerun molecular docking with selected structures. The docking results were used to analyse the possible binding pose of BV to AbPcob. A similar procedure was used for the MD simulations of AbDPcob following simulated annealing, with and RMSD cut-off of 1.0 Å, resulting in 100 clusters (15 for one binding pocket, 85 for the other) to ensure an exhaustive conformational sampling during docking, and an exhaustiveness of 25.

## Size exclusion chromatography-multi-angle light scattering (SEC-MALS)

SEC-MALS was used to determine the oligomeric state of the target proteins and to probe any protein-protein interactions, under different conditions. An Agilent G7110B HPLC pump, degasser and autoinjector (Agilient, Santa Clara, USA) was used to auto load the samples (50 μL each run, 1 to 10 mg/mL). Superdex 200 10/300 GL column was used for chromatographic separations. MiniDAWN TREOS MALS detector and Optilab rEX refractive index metre (Wyatt, Santa Barbara, USA) were used to collect light scattering signals. The flow rate was set at $1\,mL\,min^{-1}$ with a mobile phase of 20 mM HEPES, pH 6.8, 150 mM NaCl buffer. Samples were pre-incubated in dark for 30 min or with 530 nm LED for 5 min. All results were processed according to referenced protocol[53]. Peak alignment, band-broadening correction and normalisation procedures were performed by selecting the central 50% region of the peaks. Raw data were exported and plotted using Origin 9.0 software (OriginLab, Northampton, MA).

## Genomic contextual and protein-protein interaction-based enrichment analysis

A computational gene neighbourhood analysis was performed using Genomic enzymology tools[54]. Briefly, a sequence similarity network (SSN) was constructed to facilitate the assignment of function for different proteins. Then, Genome Neighbourhood Networks (GNNs) were constructed based on SSN to provide statistical analysis of neighbouring Pfam families. The frequency of common domains identified in the GNN was analysed and visualised by R and R package ggplot2. As 506 sequences belong to divergent species, four species (*Saccharothrix syringae*, *Streptomyces sp. M41*, *Micromonospora yangpuensis* and *Magnetospirillum magnetotacticum MS-1*) that possess more than one hit among 506 were selected for further analysis. In each species, the sequence was submitted to the STRING database[55] for the construction of protein-protein interaction (PPI) network using two shells (20 interactors and 5 interactors) with medium confidence (0.400). Enrichment analyses, including Gene ontology, Local network cluster (STRING), reactome pathways, protein domains, and features (Pfam and InterPro) were performed based on gene lists from the PPI network. Enrichment results were visualised using R and R package ggplot2 and ggpubr.

## Reporting summary

Further information on research design is available in the Nature Portfolio Reporting Summary linked to this article.

## Data availability

The atomic coordinates and experimental data generated in this study have been deposited in the Protein Data Bank (www.pdb.org) under accession code: 8J2W, 8J2X, 8J2Y. Atomic coordinates referenced in the main text can be found through 1B33, 3MWN. SAXS data has been deposited in the small Angle Scattering Biological Data Bank (www.sasbdb.org) under accession codes SASDUS3 and SASDUT3. LCMS data has been deposited on Figshare (Jeffreys, Laura [2024]). Photoproduct determination - LCMS. Figshare. Journal contribution. https://doi.org/10.6084/m9.figshare.25219721. LCMS data has been deposited on Figshare (Jeffreys, Laura [2024]). Photoproduct determination - NMR. Figshare. Journal contribution. https://doi.org/10.6084/m9.figshare.25219832). MD parameters for AdoCbl and OHCbl have been deposited on Figshare (https://doi.org/10.6084/m9.figshare.25226480). Initial and final structures from MD simulations have been deposited on Figshare (https://doi.org/10.6084/m9.figshare.25226549). All other source data are provided as supplementary data files. Source data are provided in this paper. Source data are provided with this paper.

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

## Acknowledgements

The work was funded by the Engineering and Physical Sciences Research Council International Centre-to-Centre grant no EP/S030336/1. We thank Diamond Light Source beamlines I03 & I04 (proposal numbers mx2447-65, 87, 89). This work was also supported by National Natural Science Foundation of China (No. 31870855) and NUDT Research Program (22-ZZCX-039, 22-02-15). The authors also thank the ESRF (proposal MX-2398) for beamtime and the staff of beamline BM29 for assistance with data collection. An ANR grant (PhotoGene, ANR-21-CE11-0036-01) awarded to GS supported the research reported in this paper. The IBS acknowledges integration into the Interdisciplinary Research Institute of Grenoble (IRIG, CEA). The authors would like to gratefully acknowledge the training and use of mass spectrometry instrumentation provided by Professor Perdita Barran's group at the Michael Barber centre for collaborative mass spectrometry (MBCCMS) based at the University of Manchester.

## Author contributions

N.S.S., D.J.H., D.L. and S.Z. initiated and coordinated the project. S.Z., L.N.J., D.J.H., N.S.S. and D.L. designed experiments, analysed data and wrote the manuscript with contributions from other authors. S.Z. produced and crystallised the proteins. H.P., M.S. and C.W.L. helped with protein crystallisation and data collection. S.Z. processed diffraction

data and solved the structures. Y.Y., L.O.J. and S.Z. performed the docking and molecular dynamics simulations. C.L. and L.Z. carried out bioinformatics analysis of Pcob proteins. K.P., L.N.J. and M.J.C. assisted with *Sas*Pcob characterisation. L.N.J. and C.Y. performed mass spectrometry analysis. L.N.J. and M.J.C. performed all NMR experiments. G.S. and M.W. performed SAXS measurements. N.S.S., D.J.H. and D.L. advised on all aspects. All authors discussed the results and commented on the manuscript.

## Competing interests

The authors declare no competing interests. Correspondence and request for materials should be addressed to N.S.S., D.J.H. and S.Z.
