## [Peer Review File · Nature Communications]

Photocobilins integrate B12 and bilin photochemistry for enzyme controlREVIEWER COMMENTS

Reviewer #1 (Remarks to the Author):

In this manuscript, Zhang et al. report the structural and spectroscopic characterization of two photoreceptors that feature two optically coupled chromophores (B12 and biliverdin) and name the protein sub-family “photocobilin”. Overall, the paper is well presented and enjoyable to read; the results appear interesting, novel, and potentially impactful. I urge the authors to keep in mind my background (computational chemistry) when reading my comments, many of which are mostly aimed at making the paper even more enjoyable and clear for a reader like me.

While the paper is in my opinion on the right track to become a great one, at times the text is a somewhat vague or “implicit”. I recommend some key modifications to enhance clarity and make the paper easy to follow for a slightly broader audience than scientists who strictly work within B12-photobiology.

1. the light-sensing/deactivation/photoconversion mechanism of B12 (adenosylcobalamin) in photoreceptors is never (briefly) described (e.g., lines 38, 44 and 81) before mention of the C-Co bond cleavage.
2. The authors often refer to “B12” throughout the text without specifying if they are referring to the adenosyl version of B12, or to cobalamins in general (e.g., R = CN, OH, CH₃..). Examples: lines 53 and 62.
3. The photocobilin family seems to be a promising new class for optogenetic applications, among others. However, the authors remain (probably intentionally) vague about what these applications may be. While I understand that these proteins just got characterized and require more study, I believe some hypotheses on what kind of existing challenges they could help address in optogenetics would help the authors make the point about novelty. In fact, the optical coupling between B12 and BV is extremely interesting and differentiates photocobilins from other B12-based photoreceptors, and the cross-talk seems to be one of the main novelty points. However, this is not, in my opinion, emphasized enough.
4. What are the wavelengths for the green and red light used at lines 74, 75 and 76, and the caption of Fig. 2? The authors could add a brief description of what the “spectral changes similar to those observed with CarH” (line 74) entail for clarity. Similarly, what “photochemical changes” are the authors referring to at line 79? I believe this is referring to the fact that the BV and AdoCbl only spectra are not precisely additive, and since this is an important point, it deserves a more extensive description. I would go as far as recommending including the extended figures 2a and 2b into the main figure 2, perhaps even in the place of current figure 2a and 2b. The B12 photoactivity triggered by BV photoabsorption is a key finding after all.
5. The “dark” and “light” state are not clearly defined (i.e., how long where the species adapted for in both cases?) on several occasions (e.g., line 79, 88 and 137). The authors don’t tell the reader why the light state of AbDPcob has not been characterized (line 89) till later in the text, leaving the reader wondering if there is a rationale.
6. How widespread are B12-photoreceptors (line 37)? Did the authors identify the two proteins they examined in the manuscript by aligning genetic code of different bacteria? These questions stem from my limited knowledge in biology, but I believe that adding a clarification on line 58 would help making

the paper understandable for the chemist audience. In other words, how did the authors know where to look to obtain these proteins?

7. I cannot see a “figure 2g and 2h” from lines 110, 130 and 134.

8. Several details for the reproducibility of the computational section are missing:

a. What version of AmberTools has been used to parametrize AdoCbl and OHCbl? How is the correct parametrization verified?

b. Is BV also parametrized using AMBER99SB force field?

c. Do the authors mean “point charges of the cofactor” at line 381 instead of “charge”? Were the charges cross-validated against different electronic structure methods and basis sets? Was the geometry optimized (and if so, which starting geometry was used, which implicit water model..)?

d. Which pH are the authors simulating? Were the protonation states given by PropKa cross-validated with other methods (e.g., H++, MCCE, CpHMD)? Were the ligands included in the PropKa calculations?

e. How many counterions were added to neutralize the simulation box, and how were the positions chosen?

f. What is the use intended for “minimize/equilibrate” (lines 389, 390, 392, 393)? Are the steps described dynamics simulations (as suggested by the length of the reported simulation steps) or energy minimizations?

g. How is the target temperature reached?

h. When positional restrains are used, are they harmonic (and if so, how strong are they) or are the coordinates completely frozen?

i. Are the hydrogen atoms constrained during the production MD (e.g., SHAKE)?

j. Was a cutoff for the electrostatic interactions used?

k. What properties were monitored to ensure that the MD simulation did not produce artifacts?

9. One of the goals of the molecular dynamic simulation section is to explore the conformational space of the proteins and investigate BV binding. Running a single long MD simulation could be considered insufficient, since, depending on the unknown phase space, the sampling might be strongly biased by the initial conditions. To make this part more robust and exhaustive, enhanced sampling techniques would be more appropriate. For instance, using temperature (or even Hamiltonian) replica exchange would help have a more comprehensive vision of the conformational space by letting the system escape from possible local minima. Such sampling could reveal several minima worth exploring with further molecular dynamics.

10. Often the authors don't properly guide the audience in what to look for in a figure to support a sentence. To give one example (of several), at lines 138-140: what is the focal point in figure 3a and 3b that the reader is supposed to focus attention on? What are the arrows pointing at? Another example: what is it in figure S2 that support the thesis expressed at line 122? Perhaps the authors could expand on the “hydrophobic interactions” mentioned at line 121, with some more guiding references to the figures and tables (for instance, the blue and red dashed lines are virtually indistinguishable).

11. Line 224: the authors may clarify what tools were used to investigate PPIs.

12. Line 310: was this achieved with the Gill and von Hippel method? A reference seems to be missing.

13. Line 315: which equation was used for which case?

FIGURES:

Fig. 1: Which one of the two photocobalins under study is depicted in this figure?

Extended data figure 1: Are the error bars determined as standard deviation from a series of

measurements? What is the x axis coordinate in panels a and b?

Fig. 2: what do the pulse numbers mean? If it is meant to represent time, it would be more useful to use the delay. What does “interaction” mean (line 552)? What is the reference for “as above” (line 553)? Is the “binding” at line 554 H-bonding?

Extended data figure 2: I suggest consistency in the coloring of the data series (e.g., the BV-only spectrum is always in red, while the AdoCbl-only and the AdoCbl+BV appear swapped in panels and B (legend typos?). Similar to above, what does the pulse # mean?

Extended data figure 3: the authors might consider zooming in with respect to the y axis in panels a-f, since the differences between the data series are very tiny and it is hard to distinguish differences (if any) in the current state. Again, pulse # meaning? What is the meaning of “350 nm \uparrow ” and “570 nm \downarrow ” in panels a and c?

Extended data figure 4: I think there is a misalignment between the caption and the figure; it appears that panels a-c have been added at a later time, and the caption has not been updated. What make a residue a “key” one (line 649)?

Extended data figure 5: what structures are shown in panels g and h to compare to the cluster ones? X-ray? How is the clustering performed (i.e., which algorithm)? Which portion of the population does each depicted cluster represent (this information would be more useful than the cluster number). If panels c-f are supposed to show that hydrophobic interactions can favor BV binding, those interactions should be shown; moreover, the reader should understand from the figure/caption that the clusters shown are the ones featuring the best binding affinities. I apologize for being perhaps paranoid, but why does the adenosyl fragment feature a bond between C5 and C8 in panel g? Is it a pymol representation issue or a parameter problem?

Figure S2: why only 500 ns out of the total 1500 are reported in panel a? Is AbPcob simulated for only 500? If so, this needs to be clarified in the computational methods section. The x axis of panel b is covered by panel d. Maybe due to the low resolution of the file, but it appears that the RMSD starts at around 2 angstroms in all cases in panels a and c; what is the reference used? I believe “Chain B” and “DGC domain” are partially covered in panel e.

Fig. 4: what does the x axis represent in panel b? If the hatch pattern represents light-adapted states, it should be clearly indicated in the legend. Is “superstition” (line 576) a typo? I am confused by the choice of the last MD frame for the comparison in panel c: why were those specific frames chosen as reference? The parallel temporal evolution argument loses a lot of importance when considering that, as mentioned above, a single MD trajectory can be strongly biased by the initial geometry and velocity. For a fair comparison, a crystal/predicted structure, and average structure or a cluster centroid would be better fitted. However, the purpose of panel c is, as far as I understand, to show key structural rearrangements in the light state. Despite the curved arrows, which I believe are meant to indicate movements, I don't think the figure in its current state is the best medium to communicate this message. Perhaps a movie? Or some zoomed-in insets for the largest changes?

Extended data figure 6: what are the units for y axes in panel a, and for the x axis in panel b? One would normally think that ticks on axes represent some kind of units, while in panel b and in other figures in the manuscript this doesn't appear to be the case.

Figure S3-4: unfortunately, these figures for me are unreadable due to both font size and resolution. I recommend increasing both.

Reviewer #2 (Remarks to the Author):

The manuscript “ Photocobilins integrate B12 and bilin photochemistry for enzyme control“ by Shaowei Zhang*, Harshwardhan Poddar, Yuqi Yu, Chuanyang Liu, Kaylee Patel, Laura N. Jeffreys, Lingyun Zhu, Cunyu Yan, Giorgio Schiro, Martin Weik, Michiyo Sakuma, Colin W. Levy, David Leys, Derren J. Heyes*, Nigel S. Scrutton*, submitted to Nature Comms, reports a combined biochemical, X-ray crystallographic, spectroscopic, computational and bioinformatics study of two representative proteins of a new apparent sub-family of multi-centre photoreceptors that bind a B12-derivative and biliverdin (BV), classified here as photocobilins.

As discovered by their crystallographic and spectroscopic work, the B12-derivative coenzyme B12 (AdoCbl) and the heme degradation product BV are positioned very closely to each other, when bound in the active site, a previously unknown arrangement, deduced to facilitate energy transfer and optical coupling. Absorption of light affects the protein structure, proposed to lead to activation of associated enzyme domains.

In one of the two photocobilins (Pcobs) analyzed, a so called stand alone photoreceptor (named SasPcob) binding of AdoCbl increases the affinity for BV decisively, whereas the other, a Pcob diguanylate cyclase fusion (named AbDPcob), binds AdoCbl well, but simultaneous binding of AdoCbl and BV is diminished compared to the situation with SasPcob.

Photocobilins are implied to be widely occurring, thus opening up a new section in the recently revealed area of B12-photobiology, and expanding decisively the scope of the whole field.

There are a number of very noteworthy properties of the here first described Pcobs, which demand a more precise data analysis and interpretation, and a thorough revision of the submission.

In fact, one major concern deals with the insufficiently characterized actual photoreaction of both of the two Pcobs. Crystallography appears to suggest the formation of ‘water/hydroxide ligated Co(III)balamin’ (lines 143-144) as one AdoCbl-photoproduct of SasPcob, but a chemical characterization of the other expected photoproduct(s) is missing. The structure of the products from the Ado-ligand would be particularly intriguing. Electron density from X-ray analysis, supporting the existence of a(n unknown) product from the Ado-ligand in the ‘light state’ (e.g.) of SasPcob, is not reported by the authors. Chromophore-protein binding studies are reported (e.g. : ext. Fig 1) on the basis of spectral changes from AdoCbl or BV binding to SasPcob - unfortunately spectra showing the observed changes are not included in this manuscript.

Remarkably, the BV forms, when bound to the two Pcobs, display characteristically different absorption spectra : are these spectra compatible with the crystal structures ? A discussion of the spectral features of the bound BV is requested, as well as of an experimental structure of the BV when bound to AbPcob. What data can be associated with the relevant excited state properties of BV in solution (e.g., energy of singlet excited state(s) above ground state) ? How would these values correlate with the corresponding values for AdoCbl ? In which direction would singlet excited state energy transfer be thermodynamically favourable ? Would the situation be qualitatively different in the protein-bound states ?

A statement and actual data concerning the extent of photoconversion(s) of the protein bound B12-derivatives (AdoCbl and MeCbl, in particular) is lacking, when BV is also bound at the active site (or not), and referring to experiments with ‘green’ light. Does the closely bound BV influence the rate of photoconversion (possibly, lower the photo-lability of the bound B12-derivative) ?

Furthermore, the statement (in the abstract, line 18) is incorrect, that 'cobalamin (vitamin B12)' can act as a light-sensing chromophore: coenzyme B12 (or adenosylcobalamin = AdoCbl) is involved here, but not vitamin B12 (cyanocobalamin).

Along these lines, an adequate description of the B12-derivatives implied would be recommended elsewhere, e.g., in the abstract, lines 23,25 and 26, as well as throughout the main parts of the manuscript.

There seems to exist a disturbing occasional lack of coherence of the figure numbers listed the manuscript and of actually presented extended figures; likewise, figure captions fail to always match actual graphics presented (look at extended figure 4, in particular).

The authors frequently refer to 'red light' or 'green light' : presumably consistently implying 530 nm and 660 nm light, respectively - if so, this should be pointed out more adequately.

The authors describe the Pcobs in a 'dark' or a 'light' state ; 'light state' is probably meant to specify a state after (not during) sufficient illumination with light ?

Describe with sufficient experimental detail how the 'light state' has been produced and how conversion of the bound B12-derivative has been monitored and made complete? What was done when handling the samples to ensure a well defined 'dark state' ?

In short, not enough detail is provided by the authors to present their interesting study with the desirable documentation of their work.

Reviewer #3 (Remarks to the Author):

The manuscript by Zhang et. al. entitled "Photocobilins integrate B12 and bilin photochemistry for enzyme control" details the structures of two photocobilins (AbPCob and SasPcob-light/dark). Through these structures, the authors identify that B12 and BV are located in close proximity and in positions that allow for optical coupling of chromophores. The structural work presented in this manuscript is interesting. However, in its current form, the novelty of the work and the insights gained from this work are not well communicated. This manuscript needs important details to be added, figures to be added, analysis to be done, and experiments to be performed.

Specific comments:

1. A reference is needed for the statement "The number of known photoreceptor families in biology has recently been expanded..."

2. Since the authors are referring to a new "subfamily" of B12-dependent proteins, the text would benefit from the inclusion of a sequence similarity network, a phylogenetic analysis, and some specific details about what families of B12-dependent proteins have been characterized. There should also be specific details regarding how many enzymes are part of the "sub-family". Apparently, the authors have an SSN based on the GNN analysis.

3. The authors should clearly make a distinction between what these proteins were previously referred to – photoglobin – and why they are now calling them – photocobilins. In reference 5, photoglobins are

identified as globin domains that are located adjacent to B12-binding domains.

4. The authors should clearly motivate their choice for the photocobilins that were studied in this work.
5. Extended Data Figure 1c does not appear to saturate and would benefit from additional measurements made at high concentrations.
6. Effort should be made to determine whether the changes are significantly different between samples illustrated in Extended data Figure 1a-b.
7. An explanation should be provided regarding why the binding ratio is above 1 in Extended data Figure 1a-b.
8. The colors and description of Extended data Figure 4a does not appear to correlate with the figure.
9. Actually, the whole legend for Extended data Figure 4 does not match the figure which has panels a-i.
10. The typical motifs for a protein to bind B12 should be clearly described in the text.
11. For the comparison of CarH and the photocobilins, there should be a figure that shows how CarH binds B12 and how the photocobilins bind B12 (page 3).
12. Extended data Figure 5 – what are conserved water molecules?
13. Page 5 – The authors describe typical linkage of BV to photoreceptor proteins but also need to provide a figure that demonstrates each of the points made for non-experts to understand how their proteins are the same or different.
14. Figure 2g does not appear to exist in the version of the manuscript that I have.
15. Figure 2h does not appear to exist in the version of the manuscript that I have.
16. Details regarding how the “light-exposed” crystal structure was determined should be added into the methods. Were the crystals or proteins exposed to light. How long did light exposure last, what wavelength of light was used, etc.
17. The differences in crystal packing between the “light-exposed” structure and dark structure should be included in the extended data.
18. The arrows in Figure 3 are not explained and it is unclear what they are showing.
19. The position of BV in Figure 3b should be indicated.
20. The fate of the cleaved Ado group should be determined for comparison to CarH (see *Biochemistry* 2015, 54, 21, 3231–3234)
21. The “severe clash” and specific movement of the residues Glu71 and His72 should be clearly illustrated.
22. Electron density maps of both B12 and BV should be included in the extended data for all structures.
23. Details regarding how the B12 and BV were refined should be added into the methods.
24. Details regarding how the kinetics assay were performed are needed – how were the concentrations determined/linear range, etc.
25. A product control should be added into the LC-MS data shown in Extended Data Figure 6a. The intensities should also be added in this figure.
26. Effort should be made to determine whether the changes are significantly different between samples illustrated in Extended data Figure 6b.
27. There should be details regarding the expression of the full length protein and efforts made to characterize its oligomeric state should be included. In addition, details should be included regarding whether the quaternary architecture is impacted by the conditions of the enzymatic assay.
28. The significant figures for the kinetic data should be consistent.
29. Based on the noted changes for MeCbl in the dark and light shown in Extended data Figure 6b, I’m confused by the authors statement (page 6) regarding “changes mediated through AdoCbl

photochemistry are important for this activation process”.

30. Page 6 – AlphaFold2 should have a reference.

31. Some effort should be made to determine whether the changes are significantly different between samples illustrated in Figure 4b.

32. It should be clearly explained whether the purification of these proteins and their incubation with B12 was performed in light or dark conditions.

33. A reference(s) should be added into the methods regarding the source of the extinction coefficients.

34. More explanation regarding the use of two different binding equations should be added into the methods.

35. Figure 4b appears to have the same data for AbDGC in the upper and lower panels but the data looks different with respect to the independent data points.

36. Figure 4b Should show additional data with higher concentrations of AbPcob until the activity is saturated.

37. There is no c label in Figure 4.

38. The arrows in panel c of Figure 4 need to be described because not all colors are described.

39. From the structures, the authors should make and test hypotheses regarding how the transition between dark and light states affects the activity of the DGC domain.

40. The illuminations assays and their analysis as does not fully support pure optical coupling (as a thermal component cannot be ruled out). Therefore, I would suggest adding temperature dependence (including lower temperatures) to the LED illumination assays to show that there is something providing energy to get from the red to the green photon. If it's not temperature dependent, the authors should provide some indication of where the energy is coming from.

41. The analysis conducted in Figures 2a/2b would benefit from SVD-based rank analysis to determine the number of contributing states (see reference 6).

42. The text (and Table 1) needs to be carefully edited for typos/extra spaces/missing spaces, etc. Some specific examples are given below.

a. Table 1 is missing (before the Rmerge of the highest resolution shell

b. Figure 2 is missing spaces before panels c and d

c. 1.7, 2.0 and 2.3 should be followed by A resolution

d. His-trap should be HisTrap (made by Cytiva)

e. Ab is not italicized in figure 4c legend.

f. Figure 4c legend light blue should be fixed

g. Page 5lines 162 and 164 appear to refer to Figure 4a not 3a.

43. The figures should be made to match the text such that the Ab or Sas are consistently italicized.

Reviewer #4 (Remarks to the Author):

Zhang et al present a very interesting study in which they characterize a novel sub-family of the B12-dependent photoreceptors related to the canonical B12 photoreceptor CarH of *Myxococcus xanthus*. In contrast to CarH, which binds adenosylcobalamine (AdoCbl) and acts as transcriptional regulator

controlling the expression of genes involved in carotenoid biosynthesis, the characterized, novel photoreceptor sub-family contain an additional globin-like domain as well as different (enzymatic active) 'effector' domains such as diguanylate cyclases (DGC). Interestingly, the proteins of the new subfamily (shown for two examples; see below) bind both AdoCbl and the linear tetrapyrrole biliverdin (BV). Given these properties, the authors coined this new subfamily of B12 photoreceptors photocobilin (photoactive-cobalamine-bilin, Pcob) proteins. While the authors are the first to experimentally characterize this family of proteins, the family was already identified recently in a bioinformatic study (cited by authors). In detail, the authors studied two proteins of the new subfamily, a 'standalone' photoreceptor lacking a fused 'effector' domain SasPcob of *Saccharothrix syringae* and a more complex multi-domain protein from an Acidimicrobiaceae bacterium containing a DGC domain as 'effector' domain (AbDPcob). The authors verified AdoCbl and BV binding in both proteins, showed that both proteins display light-dependent spectral changes similar to CarH, pointing towards a similar photochemistry. Importantly, when both AdoCbl and BV are bound to the protein, illumination with both red (BV absorption) and green (AdoCbl absorption) light, trigger photochemical changes in the B12 cofactor. In proteins loaded only with BV and illuminated with red-light, no spectral changes are observed. It thus seems that BV acts as a kind of 'antenna' to extend the absorbance range of photocobilins beyond the blue/green region of the spectrum. In addition to detailed spectroscopic studies, the authors also obtained crystal structures of the dark and light state of SasPcob and the AdDPcob photoreceptor domain (without DGC), which provided insight into AdoCbl and BV binding. Since the authors could not obtain a crystal structure of the full-length AbDPcob protein, SAXS was used to study the solution structure of the dark and light state of the protein, revealing significant structural differences. Using the full-length AdDPcob protein, light-dependent DGC activity was verified for the full-length protein. Interestingly, when the isolated DGC domain of AdDPcob was mixed with the isolated photocobilin photoreceptor domains (AdPcob and SasPcob) a light dependent change in DGC activity, which was concentration dependent, was observed, suggesting that in the absence of a fused effector domain standalone photocobilin could control effector activity via protein-protein interactions. Last but not least, bioinformatic studies (e.g. genome proximity analyses) were used to elucidate the potential functions of photocobilin photoreceptors.

The presented work is novel, original and of general interest for the broader biology/microbiology/biophysics community. The work is carried out well, the paper is generally well written and the conclusions are sound. I thus believe the manuscript deserves publication. However, I have a few (rather minor) comments that the authors should consider to improve broader readability beyond the more specialized audience of photobiologists and to place their conclusions on more solid ground.

Major comments;

I found reading the manuscript rather difficult due to the differentiation between supporting information and extended data. Is there a reason for not placing the extended data into a single supporting information file?

A brief introduction in CarH photochemistry would be needed for readers that are not from the field, i.e. to understand the conclusions drawn by the authors (see page 3, line 98. Discussion about 'His-on ligation')

Given the fact, that nothing is known about the natural functions of the photocobin photoreceptors, can the authors be sure that BV is the natural second chromophore of these proteins? Did the authors test binding of PCB, PEB etc?

Along those lines, is anything known about bilin synthesis in the host microbes *Saccharothrix syringae* and *Acidimicrobiaceae* bacterium?

The authors experimentally show that excitation of the BV and AdoCbl loaded proteins at a wavelength where only BV absorbs results in spectral changes resulting from photochemistry of the B12 chromophore, which suggests that BV can act as a type of 'antenna' pigment somewhat reminiscent of the MTHF / FAD system in CPD photolyases and Cry-DASH cryptochromes. However, in those latter systems antenna function is explained by FRET from MTHF to FAD. I don't see how 'antenna' function might be realized in the BV/AdoCbl system. Can the authors speculate how excitation of BV can trigger a CarH like AdoCbl photochemistry?

Please clarify how dark and light state structures were obtained? Did the authors crystallize the proteins in the dark and collect data from dark and illuminated crystals or was the light state preformed and crystallized separately. From reading the methods text 'Crystallisation was performed in the dark using the sitting drop vapour diffusion technique ...' it sounds like the first scenario. Here, on page 3-5 the authors state that the Pcob photoreceptor domain is dimeric (page 3, line 95) in the dark and a monomer in the light state (see page 5, line 139) and that illumination triggers monomerization. I have two questions in this regard:

i) How is such an assessment possible when the proteins were crystallized in the dark? Wouldn't a large-scale structural change such as monomerization break the crystal lattice?

ii) Is there further experimental proof for this assessment e.g. from SAXS studies of the isolated Pcob photoreceptor domains or SEC-MALS? If not, some further proof would be needed.

SAXS data assessment should be improved: The authors recorded SAXS data for the dark and light state of AbDPcob and performed MD simulations of the protein in both states. On page 6 line 187/188 the authors state that 'AlphaFold2 prediction for AbDPcob is compatible with the dark conformation in solution. From the Figure presented in Extended Data set 6, it appears that this conclusion is based on the fact that the full-length model can be fitted into the DAMMIF generated ab initio 'envelope'. To quantify this assessment, the authors could use CRY SOL to 'back-calculate' SAXS data from the AlphaFold2 model and fit this data against the experimental data. This can also be done for selected snapshots from the light state MD simulation to see which structure best fits the experimental data. In this way, the authors might get some information about global light-dependent structural changes.

Apart from genome proximity analyses, some words about the distribution of putative photocobins would be warranted. In which bacteria are they found? Are there any common physiological traits to these organisms?

Minor comments

Page 5, line 140: light-triggered departure of the ... is there a better word for departure? Light-triggered release? photolysis?

Page 5, line 162 and line 164: isn't this Figure 4a and not 3a?

Extended Data Figure 1 and 2 legend: What are AbBGD and SasBG? = AbDPcob and SasPcob?

Extended Data Figure 4 legend: Something doesn't fit here? The figure legend does not seem to match what is shown in the figure. Please clarify!

REVIEWER COMMENTS

Reviewer 1:

In this manuscript, Zhang et al. report the structural and spectroscopic characterization of two photoreceptors that feature two optically coupled chromophores (B12 and biliverdin) and name the protein sub-family “photocobilin”. Overall, the paper is well presented and enjoyable to read; the results appear interesting, novel, and potentially impactful. I urge the authors to keep in mind my background (computational chemistry) when reading my comments, many of which are mostly aimed at making the paper even more enjoyable and clear for a reader like me.

While the paper is in my opinion on the right track to become a great one, at times the text is a somewhat vague or “implicit”. I recommend some key modifications to enhance clarity and make the paper easy to follow for a slightly broader audience than scientists who strictly work within B12-photobiology.

We would like to thank the referee for their complimentary comments on our work. We are pleased that the reviewer feels our work is potentially impactful and have now addressed the points raised, which hopefully make the paper easier to follow for a broader audience.

1. the light-sensing/deactivation/photoconversion mechanism of B12 (adenosylcobalamin) in photoreceptors is never (briefly) described (e.g., lines 38, 44 and 81) before mention of the C-Co bond cleavage.

We have added a few lines to clarify the photochemical mechanism of adenosylcobalamin photoreceptors in general. Please see the additional text in lines 46-49.

2. The authors often refer to “B12” throughout the text without specifying if they are referring to the adenosyl version of B12, or to cobalamins in general (e.g., R = CN, OH, CH₃..). Examples: lines 53 and 62.

We use B₁₂ for the general representation of all B₁₂s. However, as it is only adenosylcobalamin that is used by such photoreceptors we have now used this terminology throughout to clarify which B₁₂ form we are referring to.

3. The photocobilin family seems to be a promising new class for optogenetic applications, among others. However, the authors remain (probably intentionally) vague about what these applications may be. While I understand that these proteins just got characterized and require more study, I believe some hypotheses on what kind of existing challenges they could help address in optogenetics would help the authors make the point about novelty. In fact, the optical coupling between B12 and BV is extremely interesting and differentiates photocobilins from other B12-based photoreceptors, and the cross-talk seems to be one of the main novelty points. However, this is not, in my opinion, emphasized enough.

We have included further details about how Pcobs could potentially be used in optogenetics (see lines 51-56) In particular, AdoCbl-dependent photoreceptors have shown great promise in light-dependent hydrogels for drug delivery but are often limited by the lack of penetration of

green light through the skin. The Pcob family could potentially be transformative as it would allow the same hydrogel technology to be sensitive to the more penetrative red / far-red light. We agree with the reviewer that the cross talk between B₁₂ and BV is one of the main novelty points and we have tried to emphasise this more in the manuscript. However, the nature of this cross-talk is likely to be complex and is not currently fully understood. Hence, we are still investigating the mechanism of how the cofactors ‘talk’ to each other.

4. What are the wavelengths for the green and red light used at lines 74, 75 and 76, and the caption of Fig. 2? The authors could add a brief description of what the “spectral changes similar to those observed with CarH” (line 74) entail for clarity. Similarly, what “photochemical changes” are the authors referring to at line 79? I believe this is referring to the fact that the BV and AdoCbl only spectra are not precisely additive, and since this is an important point, it deserves a more extensive description. I would go as far as recommending including the extended figures 2a and 2b into the main figure 2, perhaps even in the place of current figure 2a and 2b. The B₁₂ photoactivity triggered by BV photoabsorption is a key finding after all.

The green and red light wavelengths used are 530 nm and 660 nm respectively. This has now been clarified and added to the main text. A brief description of the spectral changes has been added into the main text on lines 88-90. As suggested, we have also now included the extended data figures 2a and 2b in the main figure 2.

5. The “dark” and “light” state are not clearly defined (i.e., how long where the species adapted for in both cases?) on several occasions (e.g., line 79, 88 and 137). The authors don’t tell the reader why the light state of AbDPcob has not been characterized (line 89) till later in the text, leaving the reader wondering if there is a rationale.

The details for “dark” and “light” state have been added in the method section (see page 10, line 305-309). The reason why AbDPcob has not been characterised has been added to the text on page 4, line 106-107.

6. How widespread are B₁₂-photoreceptors (line 37)? Did the authors identify the two proteins they examined in the manuscript by aligning genetic code of different bacteria? These questions stem from my limited knowledge in biology, but I believe that adding a clarification on line 58 would help making the paper understandable for the chemist audience. In other words, how did the authors know where to look to obtain these proteins?

Bioinformatics studies on B₁₂ photoreceptors have already been published in refs 5 and 6, which identified these proteins as well as several others. We have clarified this on line 59-60 and included further bioinformatics analyses about Pcob distribution in supplementary information figures 15-19.

7. I cannot see a “figure 2g and 2h” from lines 110, 130 and 134.

We apologise for this error. It has now it has been updated to figure 2e and 2f. Please also see line 116, line 120 and line 153.

8. Several details for the reproducibility of the computational section are missing:
a. What version of AmberTools has been used to parametrize AdoCbl and OHCbl? How is the correct parametrization verified?

AmberTools 22 was used to parameterise AdoCbl and OHCbl. We have added this to the main text. See page 13, line 422-423.

We use MCPB.py protocol to prepare the input files for Gaussian to calculate force constant and charges for the two ligands. This is commonly used to prepare parameters for metalloproteins, and it has been proved to work well on metalloprotein by other studies. The structural stability of the ligands and ligand-binding across the simulations suggests that parametrisation was adequate.

b. Is BV also parametrized using AMBER99SB force field?

No. We did not include BV in the simulation. MD was mainly run for AbDPcob sample, which was not crystallised with BV due to the binding issue.

c. Do the authors mean “point charges of the cofactor” at line 381 instead of “charge”? Were the charges cross-validated against different electronic structure methods and basis sets? Was the geometry optimized (and if so, which starting geometry was used, which implicit water model..)?

For metal ions, the DFT functional B3LYP with the 6-31G* basis set has been commonly used for RESP charge evaluation because of its overall accuracy and speed. The ligand from the crystal structure was used as the starting geometry and then it was optimised by Gaussian. The optimised structure was used for calculating point charges and force constant. We validated the forcefield of the ligand by monitoring its geometry after optimisation and during the dynamic simulation (see the response for 8a). We did not cross-validate against different methods and basis sets.

d. Which pH are the authors simulating? Were the protonation states given by PropKa cross-validated with other methods (e.g., H++, MCCE, CpHMD)? Were the ligands included in the PropKa calculations?

We are simulating at the neutral pH 7.0 and the B₁₂ was included in the Propka calculations. This has been added to the main text. We did not cross-validate the PropKa results as we have generally found that PropKa is adequate for determining the protonation states, and since the MD is very stable with minimal structural changes over both the long simulations and the new simulations following simulated annealing, this suggests that there were no significant errors in the protonation states. Additionally, the BV binding site is largely hydrophobic so the pKa of the immediate surrounding residues are unlikely to be crucial.

e. How many counterions were added to neutralize the simulation box, and how were the positions chosen?

The counterions were added by using Leap module in AmberTools 22 in a shell around the structure and the positions were calculated based on a Coulombic potential to minimise the electrostatic energy of the system. 19 Na⁺ ions were added for truncated *AbPcob* and 58 Na⁺ ions were for full-length *AbDPcob*. See page 13, line 434-438.

f. What is the use intended for “minimize/equilibrate” (lines 389, 390, 392, 393)? Are the steps

described dynamics simulations (as suggested by the length of the reported simulation steps) or energy minimizations?

The system was first minimised to remove possible clashes after adding hydrogens to the protein, followed by a heating step to 300 K under NVT ensemble. The stepwise equilibration steps are conducted in order to make sure the structure is gradually relaxed in the solvent environment. These procedures are only in preparation to facilitate the subsequent dynamics simulation. Therefore, the length of the reported simulations did not include the previous steps and only contain the dynamics after the structure fully relaxed in the solvent. The Methods section has been updated to address the point. See page 13, lines 434-438.

g. How is the target temperature reached?

In the MD simulation the temperature is calculated based on the kinetic energy of the system which is related to velocities of atoms. The initial velocities were generated to maintain a Maxwell-Boltzmann distribution at the desired temperature, 300 K. To control the temperature, we use the V-rescale thermostat algorithm.

h. When positional restrains are used, are they harmonic (and if so, how strong are they) or are the coordinates completely frozen?

Harmonic restraint was added before full relaxation of the structures with a force constant of 1000 kJ/mol/nm². We have added this to the main text. See page 14, lines 444-445.

i. Are the hydrogen atoms constrained during the production MD (e.g., SHAKE)?

Yes, the hydrogen atoms were constrained during the production by LINCS. We have added this to the main text. See page 14, lines 449-450.

j. Was a cutoff for the electrostatic interactions used?

The cut-off was set to 12 Å for electrostatic interactions calculation. We have added this to the main text. See page 14, line 450.

k. What properties were monitored to ensure that the MD simulation did not produce artifacts?

As shown in SI Figure S6 the C α RMSD is stable, and we observe no significant structural changes over the course of the simulations. Additionally, the structures were found to be stable during and after 3 \times 50 ns simulated annealing runs (SI Figure S7), despite much larger C α RMSDs. This suggests that the MD simulations are not showing any significant artifacts.

9. One of the goals of the molecular dynamic simulation section is to explore the conformational space of the proteins and investigate BV binding. Running a single long MD simulation could be considered insufficient, since, depending on the unknown phase space, the sampling might be strongly biased by the initial conditions. To make this part more robust and exhaustive, enhanced sampling techniques would be more appropriate. For instance, using temperature (or even Hamiltonian) replica exchange would help have a more comprehensive vision of the conformational space by letting the system escape from possible local minima. Such sampling could reveal several minima worth exploring with further molecular dynamics.

The reviewer is correct that a single MD run is not sufficient for exhaustive conformational sampling, as the system can get stuck in a local minimum (metastable state). However, in this case the aims were to determine whether thermal fluctuations would be sufficient to open up the binding pocket so that BV could bind, and whether the dynamics of the dark and light states are different in the current conformation. Replica exchange would be computationally too expensive in this case, due to the large number of replicas required for such a large system. However, we have performed an additional series of simulations in parallel, each starting with 50 ns of simulated annealing (described in the Methods). As shown in SI Figure S7, this has dramatically increased the degree of conformational sampling while the structure remains stable (SI Figure S7). From this, we were able to dock BV in a structure that represents a much larger cluster than previously (SI Figure S8).

10. Often the authors don't properly guide the audience in what to look for in a figure to support a sentence. To give one example (of several), at lines 138-140: what is the focal point in figure 3a and 3b that the reader is supposed to focus attention on? What are the arrows pointing at?

We apologise for not referring to the figure properly. We have now rearranged all the figures to hopefully guide the reader in a clearer manner. For figure 3a and 3b (now figure 7d-7e), they show the overall structural changes of CarH and *SasPcob*, the arrow indicates the magnitude of the movement of the protein. This has been clarified in the figure legend (see page 21, line 730-731).

Another example: what is it in figure S2 that support the thesis expressed at line 122? Perhaps the authors could expand on the "hydrophobic interactions" mentioned at line 121, with some more guiding references to the figures and tables (for instance, the blue and red dashed lines are virtually indistinguishable).

In figure S2 (now figure S5), panels a and b show parameters from the MD simulation for *AbPcob* in light and dark state to illustrate that there was no obvious movement of the protein. We chose the dark state of *AbPcob* to explore the possible binding of BV to this protein. We have included more description of the hydrophobic interactions and included more text (see page 5, lines 139-145) to guide the reader around these findings.

11. Line 224: the authors may clarify what tools were used to investigate PPIs.

The STRING database was used for the PPI prediction and is clarified in the method section on page 15, lines 495-497.

12. Line 310: was this achieved with the Gill and von Hippel method? A reference seems to be missing.

The protein extinction coefficients were calculated using the ProtParam tool (<https://web.expasy.org/protparam/>). A reference has been added in the method section (Gasteiger, E. et al. in *The Proteomics Protocols Handbook* (ed John M. Walker) 571-607 (Humana Press, 2005)).

13. Line 315: which equation was used for which case?

The equation used for binding analysis has been clarified in the method section. See page 11, line 351-353.

FIGURES:

Fig. 1: Which one of the two photocobalins under study is depicted in this figure?

Figure 1 shows the spectra and structure of *SasPcob* and this has now been clarified in the figure legend.

Extended data figure 1: Are the error bars determined as standard deviation from a series of measurements? What is the x-axis coordinate in panels a and b?

This is now figure 3 in the revised version of the manuscript. The error bars were determined by the standard deviation from triplicate measurements. The x-axis in panels a and b correspond to different samples, which have been labelled in the figure.

Fig. 2: What do the pulse numbers mean? If it is meant to represent time, it would be more useful to use the delay. What does “interaction” mean (line 552)? What is the reference for “as above” (line 553)? Is the “binding” at line 554 H-bonding?

We used 100 ms light pulses to illuminate the samples and this has now been clarified in the figure. The term ‘interaction’ here describes how the chromophore molecules bind to or interact with the *SasPcob* protein. The term “as above” has been clarified in the figure legend.

Extended data figure 2: I suggest consistency in the coloring of the data series (e.g., the BV-only spectrum is always in red, while the AdoCbl-only and the AdoCbl+BV appear swapped in panels A and B (legend typos?). Similar to above, what does the pulse # mean?

Thanks for the suggestion. The colour has been changed to keep the consistency. “Pulse” has been explained as above and has now been clarified in the figure.

Extended data figure 3: The authors might consider zooming in with respect to the y-axis in panels a-f, since the differences between the data series are very tiny and it is hard to distinguish differences (if any) in the current state. Again, pulse # meaning? What is the meaning of “350 nm \uparrow ” and “570 nm \downarrow ” in panels a and c?

Thanks for the suggestion. We have modified the figure as suggested and have zoomed in to make the specific area clearer. The “ \uparrow ” is used to illustrate there is an increase in the absorbance at 350 nm, whilst the “ \downarrow ” shows a decrease at 570 nm. Please see page 19, line 682-683.

Extended data figure 4: I think there is a misalignment between the caption and the figure; it appears that panels a-c have been added at a later time, and the caption has not been updated. What makes a residue a “key” one (line 649)?

We apologise for this oversight. The figure legends have been corrected in what is now figure 5 (see page 19). Key residues here mean the main residues involved or around dimer interface and this has been clarified in the figure legend.

Extended data figure 5: What structures are shown in panels g and h to compare to the cluster ones? X-ray?

This is now figure 6 in the revised version of the manuscript. Panel g is the BV docked model of *AbPcob* and panel h is the crystal structure of *SasPcob*. It has been clarified in the figure legend (see page 20, line 716).

How is the clustering performed (i.e., which algorithm)?

After 500 ns MD simulation, the *AbPcob* dark structure was clustered using Gromacs gromos method, which uses an algorithm as described in Daura et al. (*Angew. Chem. Int. Ed.* 1999, 38, pp 236-240). To count number of neighbours using cut-off at 1.5 angstrom, structure with largest number of neighbours was taken with all its neighbours as cluster and eliminated it from the pool of clusters. More details have been included in method section (see page 14, line 460).

Which portion of the population does each depicted cluster represent (this information would be more useful than the cluster number).

A table has been included in the SI (supplementary table 2) to explain the relation of cluster number and structure populations.

If panels c-f are supposed to show that hydrophobic interactions can favor BV binding, those interactions should be shown; moreover, the reader should understand from the figure/caption that the clusters shown are the ones featuring the best binding affinities. I apologize for being perhaps paranoid, but why does the adenosyl fragment feature a bond between C5 and C8 in panel g? Is it a pymol representation issue or a parameter problem?

Hydrophobic interaction has been added in the figure (now figure 6c-6f). The bond between C5 and C8 is a Pymol representation issue. This has now been fixed in the new figure (figure 6g).

Figure S2: why only 500 ns out of the total 1500 are reported in panel a? Is *AbPcob* simulated for only 500? If so, this needs to be clarified in the computational methods section. The x axis of panel b is covered by panel d. Maybe due to the low resolution of the file, but it appears that the RMSD starts at around 2 angstroms in all cases in panels a and c; what is the reference used? I believe “Chain B” and “DGC domain” are partially covered in panel e.

AbPcob is only simulated for 500 ns. It has been clarified in the method section (see page 13, line 446-447). We apologise for the issues in the figure. It has been modified to fix them (now Figure S6). RMSD value was calculated with crystal structure as the reference structure.

Fig. 4: what does the x axis represent in panel b? If the hatch pattern represents light-adapted states, it should be clearly indicated in the legend.

The x axis in panel b represents different samples, which has already been labelled in the figure. The hatch pattern is for light-adapted states. The figure legend has been modified to address this point, see updated figure 8.

Is “superstition” (line 576) a typo? I am confused by the choice of the last MD frame for the comparison in panel c: why were those specific frames chosen as reference? The parallel temporal evolution argument loses a lot of importance when considering that, as mentioned above, a single MD trajectory can be strongly biased by the initial geometry and velocity. For

a fair comparison, a crystal/predicted structure, and average structure or a cluster centroid would be better fitted.

We apologise for the typo. It should be “superposition”. Sorry for misleading about the MD “frame”. The MD results were clustered and represent structure were integrate to multiple states to show the trajectory and protein movement. Among them, starting and final structures were chosen for making figures.

However, the purpose of panel c is, as far as I understand, to show key structural rearrangements in the light state. Despite the curved arrows, which I believe are meant to indicate movements, I don't think the figure in its current state is the best medium to communicate this message. Perhaps a movie? Or some zoomed-in insets for the largest changes?

Thanks for the suggestion. Movies have been made to show the protein movement. See supplementary movies 1 and 2.

Extended data figure 6: what are the units for y axes in panel a, and for the x axis in panel b? One would normally think that ticks on axes represent some kind of units, while in panel b and in other figures in the manuscript this doesn't appear to be the case.

The y axis in panel a is the signal intensity of the mass spec data. The units and values have been added in the figure (updated figure 9). The x axis in panel b corresponds to different samples, which have already been labelled in the figure.

Figure S3-4: unfortunately, these figures for me are unreadable due to both font size and resolution. I recommend increasing both.

We apologise for the low resolution of the original figure and have improved the resolution in the revised version. We have also included these as separate figures (please see SI figures 16-18).

Reviewer 2:

The manuscript “Photocobilins integrate B12 and bilin photochemistry for enzyme control“ by Shaowei Zhang*, Harshwardhan Poddar, Yuqi Yu, Chuanyang Liu, Kaylee Patel, Laura N. Jeffreys, Lingyun Zhu, Cunyu Yan, Giorgio Schiro, Martin Weik, Michiyo Sakuma, Colin W. Levy, David Leys, Derren J. Heyes*, Nigel S. Scrutton*, submitted to Nature Comms, reports a combined biochemical, X-ray crystallographic, spectroscopic, computational and bioinformatics study of two representative proteins of a new apparent sub-family of multi-centre photoreceptors that bind a B12-derivative and biliverdin (BV), classified here as photocobilins.

As discovered by their crystallographic and spectroscopic work, the B12-derivative coenzyme B12 (AdoCbl) and the heme degradation product BV are positioned very closely to each other, when bound in the active site, a previously unknown arrangement, deduced to facilitate energy transfer and optical coupling. Absorption of light affects the protein structure, proposed to lead to activation of associated enzyme domains.

In one of the two photocobilins (Pcobs) analyzed, a so called stand alone photoreceptor (named SasPcob) binding of AdoCbl increases the affinity for BV decisively, whereas the other, a Pcob diguanylate cyclase fusion (named AbDPcob), binds AdoCbl well, but simultaneous binding of AdoCbl and BV is diminished compared to the situation with SasPcob.

Photocobilins are implied to be widely occurring, thus opening up a new section in the recently revealed area of B12-photobiology, and expanding decisively the scope of the whole field.

There are a number of very noteworthy properties of the here first described Pcobs, which demand a more precise data analysis and interpretation, and a thorough revision of the submission.

We would like to thank the referee for their positive comments on our work and are pleased they feel our work '*could expand decisively the scope of the whole field*'. We have now revised the manuscript accordingly to provide a more precise analysis and interpretation of the data.

In fact, one major concern deals with the insufficiently characterized actual photoreaction of both of the two Pcobs. Crystallography appears to suggest the formation of 'water/hydroxide ligated Co(III)balamin' (lines 143-144) as one AdoCbl-photoproduct of SasPcob, but a chemical characterization of the other expected photoproduct(s) is missing.

We have now used both native MS, LC-MS and NMR to characterise the other photoproduct and this has been identified as 4', 5' -anhydro-adenosine. This is the same photoproduct that was previously observed for the well-studied CarH system. The data is included as figures S9-S10.

The structure of the products from the Ado-ligand would be particularly intriguing. Electron density from X-ray analysis, supporting the existence of a(n unknown) product from the Ado-ligand in the 'light state' (e.g.) of SasPcob, is not reported by the authors.

As the Ado ligand leaves the B12 binding pocket entirely upon illumination it cannot be observed by X-ray crystallography in the light state. Hence, as described above, this is why we have now identified this chemically by LC-MS methods.

Chromophore-protein binding studies are reported (e.g. : ext. Fig 1) on the basis of spectral changes from AdoCbl or BV binding to SasPcob - unfortunately spectra showing the observed changes are not included in this manuscript.

We have now included these raw spectra changes upon cofactor binding in SI figure S3.

Remarkably, the BV forms, when bound to the two Pcobs, display characteristically different absorption spectra : are these spectra compatible with the crystal structures ? A discussion of the spectral features of the bound BV is requested, as well as of an experimental structure of the BV when bound to AbPcob.

We have now added a few lines to discuss the BV binding features (see page 5, lines 143-145) As shown in the manuscript, the BV does not bind as efficiently to *AbPcob* as it does to *SasPcob*, with only around 20 % incorporation. The key binding interaction to Asp270 in *SasPcob* is missing in *AbPcob*, which likely explains the difference in BV binding and the associated spectral changes observed. As a result of this weaker BV binding in *AbPcob* we failed to obtain a *AbPcob* structure with BV bound.

What data can be associated with the relevant excited state properties of BV in solution (e.g., energy of singlet excited state(s) above ground state) ? How would these values correlate with the corresponding values for AdoCbl ? In which direction would singlet excited state energy

transfer be thermodynamically favourable ? Would the situation be qualitatively different in the protein-bound states ?

This is an excellent point and one that will require a lot of further investigation using a range of biophysical spectroscopy and computational chemistry techniques. Intuitively, one would imagine that excitation of BV with red light would mean that singlet state energy transfer to the B₁₂ is unfavourable. Hence, the mechanism of coupling between the 2 cofactors is currently not clear and indeed, may occur via a different route (e.g. triplet energy transfer, electron transfer, etc). Due to this uncertainty, we have been careful not to state what the mechanism might be and just claim that they are optically coupled to avoid any possible confusion.

A statement and actual data concerning the extent of photoconversion(s) of the protein bound B₁₂-derivatives (AdoCbl and MeCbl, in particular) is lacking, when BV is also bound at the active site (or not), and referring to experiments with 'green' light. Does the closely bound BV influence the rate of photoconversion (possibly, lower the photo-lability of the bound B₁₂-derivative) ?

This is a good point but is difficult to measure experimentally. Although the BV cofactor predominantly absorbs red light it also has significant absorbance across the entire visible spectrum. Hence, when samples with both BV and B₁₂ bound to the protein are illuminated with green light you get a mixture of direct B₁₂ excitation and the BV-coupled excitation route. It is impossible to disentangle the 2 processes.

Furthermore, the statement (in the abstract, line 18) is incorrect, that 'cobalamin (vitamin B₁₂)' can act as a light-sensing chromophore: coenzyme B₁₂ (or adenosylcobalamin = AdoCbl) is involved here, but not vitamin B₁₂ (cyanocobalamin).

We have changed this throughout the manuscript to AdoCbl to mean photolabile form of B₁₂.

Along these lines, an adequate description of the B₁₂-derivatives implied would be recommended elsewhere, e.g., in the abstract, lines 23,25 and 26, as well as throughout the main parts of the manuscript.

We have added a description of B₁₂ derivatives in the introduction (page 2, lines 38-41).

There seems to exist a disturbing occasional lack of coherence of the figure numbers listed the manuscript and of actually presented extended figures; likewise, figure captions fail to always match actual graphics presented (look at extended figure 4, in particular).

We apologise for the consistency in figure numbering. We have now reworked all the figures to make a more coherent story for the reader.

The authors frequently refer to 'red light' or 'green light' : presumably consistently implying 530 nm and 660 nm light, respectively - if so, this should be pointed out more adequately.

We have clarified that green and red light refer to 530 nm and 660 nm, respectively in the main text (see page 3, lines 86-88 and page 10, line 337-338).

The authors describe the Pcobs in a 'dark' or a 'light' state ; 'light state' is probably meant to specify a state after (not during) sufficient illumination with light ?

The light state means sample that has been fully converted to a light-adapted state following sufficient illumination with light. It has been clarified in the main text (see page 4, line 103).

Describe with sufficient experimental detail how the ‘light state’ has been produced and how conversion of the bound B12-derivative has been monitored and made complete? What was done when handling the samples to ensure a well defined ‘dark state’ ?

The light state sample was prepared under 530nm LED (or room light) for at least 5 mins. The illuminated sample was checked by absorbance spectroscopy to ensure that it had fully photoconverted. We process the dark samples under dimmed red light, which doesn’t cause any photoconversion of the sample. More details of this preparation of the samples have been added in the method section (see pages 10, lines 311-315).

In short, not enough detail is provided by the authors to present their interesting study with the desirable documentation of their work.

Thanks for all the good suggestions. We have added all of the requested information to the main text or method section.

Reviewer 3:

The manuscript by Zhang et. al. entitled “Photocobilins integrate B12 and bilin photochemistry for enzyme control” details the structures of two photocobilins (AbPCob and SasPcob-light/dark). Through these structures, the authors identify that B12 and BV are located in close proximity and in positions that allow for optical coupling of chromophores. The structural work presented in this manuscript is interesting. However, in its current form, the novelty of the work and the insights gained from this work are not well communicated. This manuscript needs important details to be added, figures to be added, analysis to be done, and experiments to be performed.

We are pleased that the referee feels our work is interesting and have now revised the manuscript to hopefully communicate our findings in an improved manner.

Specific comments:

1. A reference is needed for the statement “The number of known photoreceptor families in biology has recently been expanded...”

Reference 2 has been added to the statement (see page 2 and 15, reference 2).

2. Since the authors are referring to a new “subfamily” of B12-dependent proteins, the text would benefit from the inclusion of a sequence similarity network, a phylogenetic analysis, and some specific details about what families of B12-dependent proteins have been characterized. There should also be specific details regarding how many enzymes are part of the “sub-family”. Apparently, the authors have an SSN based on the GNN analysis.

Sequence similarity network and phylogenetic analysis were carried out based on the *SasPcob* protein sequence. The SSN results demonstrated five distinct clusters, with *SasPcob*, *CarH*, and methionine synthase grouped together in one cluster. Phylogenetic analysis indicate that *Pcob*

proteins are widely distributed across various types of microorganisms, ranging from archaea to bacteria. For more details please see supplementary results section (SI Figures S13-S20).

3. The authors should clearly make a distinction between what these proteins were previously referred to – photoglobin – and why they are now calling them – photocobilins. In reference 5, photoglobins are identified as globin domains that are located adjacent to B₁₂-binding domains.

The term ‘photoglobin’ was used by Schneider, T. *et.al.* (2022) to describe the globin-like domain that are widespread in nature. Figure 4 in this original paper shows that although the photoglobin domains are often located adjacent to a B₁₂-binding domain they are also found in conjunction with a wide array of other domains. As we now show that these domains bind bilin rather than heme we have used the term photocobilin (photo-cobalmin-bilin) to describe the fusion of these photoglobin domains with the neighbouring B₁₂-binding domain. We have included an extra line (see page 8, lines 245-246) to clarify this.

4. The authors should clearly motivate their choice for the photocobilins that were studied in this work.

We have selected 2 types of photocobilins to study, one that represents a standalone photocobilin photoreceptor and another that is representative of a more complex photocobilin domain fused to an enzyme domain. We have tried to clarify this in the revised version of the manuscript (see page 3, lines 70-73).

5. Extended Data Figure 1c does not appear to saturate and would benefit from additional measurements made at high concentrations.

In these experiments (now updated to be figure 3) we are limited by the protein concentration that we are able to use as this is what we titrate in to measure binding. The *apo* protein is not stable at higher concentrations but the data fits well to illustrate the point that binding is weaker in the absence of BV.

6. Effort should be made to determine whether the changes are significantly different between samples illustrated in Extended data Figure 1a-b.

A statistical analysis has been carried out for panel a and b (please see updated figure 3).

7. An explanation should be provided regarding why the binding ratio is above 1 in Extended data Figure 1a-b.

The concentration of chromophore and protein were calculated based on their absorbance and extinction coefficient. However, for BV binding it is very difficult to get an accurate value as the absorbance increases upon binding (see figure 3f and figure S3). Hence, this may explain why the binding ratio is slightly higher than 1 in this case.

8. The colors and description of Extended data Figure 4a does not appear to correlate with the figure.

We apologise for this oversight in the figure legend and now corrected accordingly (please see new figure 5).

9. Actually, the whole legend for Extended data Figure 4 does not match the figure which has panels a-i.

Again, we apologise for this mistake and have corrected it in the revised version (please see new figure 5).

10. The typical motifs for a protein to bind B12 should be clearly described in the text.

We have added a few lines to describe the B₁₂ binding motif found in other B₁₂ photoreceptors (see page 4, lines 109-111).

11. For the comparison of CarH and the photocobilins, there should be a figure that shows how CarH binds B12 and how the photocobilins bind B12 (page 3).

We have modified figure 7 to illustrate the point (see figure 7, panels a-c).

12. Extended data Figure 5 – what are conserved water molecules?

The conserved water molecules in updated figure 6 refer to water molecules that appear in several crystal data collection sets and therefore appear to be structurally important. We have changed the term ‘conserved’ to ‘structurally-relevant’.

13. Page 5 – The authors describe typical linkage of BV to photoreceptor proteins but also need to provide a figure that demonstrates each of the points made for non-experts to understand how their proteins are the same or different.

Thanks for the suggestion. Figure 5 (panels g and h) has now been modified to address this point showing a comparison of Pcob to other BV photoreceptors. It has also been clarified in the main text (see page 4, line 127-133).

14. Figure 2g does not appear to exist in the version of the manuscript that I have.

15. Figure 2h does not appear to exist in the version of the manuscript that I have.

We apologise for this oversight and have corrected the text to refer to figures 2e and 2f.

16. Details regarding how the “light-exposed” crystal structure was determined should be added into the methods. Were the crystals or proteins exposed to light. How long did light exposure last, what wavelength of light was used, etc.

We have added the details for obtaining the light crystals. Please see method section, page 10, lines 311-315.

17. The differences in crystal packing between the “light-exposed” structure and dark structure should be included in the extended data.

Thanks for the suggestion. Crystal packing for dark and light models has been shown in figure S5d-S5e.

18. The arrows in Figure 3 are not explained and it is unclear what they are showing.

The figure legends have now been modified to clarify the issue (please see updated Figure 7). The arrow indicates the direction in which the protein moves, while the length of the arrow represents the scale of the movement.

19. The position of BV in Figure 3b should be indicated.

Thanks for the suggestion. The position of BV in figure 3b (now figure 7e) has been indicated by a grey sphere.

20. The fate of the cleaved Ado group should be determined for comparison to CarH (see *Biochemistry* 2015, 54, 21, 3231–3234)

We have now used both native MS, LC-MS and NMR to characterise the other photoproduct and this has been identified as 4', 5' anhydroadenosine. This is the same photoproduct that was previously observed for the well-studied CarH system in the reference that the reviewer is referring to. This data is included as figures S9-S10.

21. The “severe clash” and specific movement of the residues Glu71 and His72 should be clearly illustrated.

We have modified Figure 7 to address the point. Residues movement and its potential clash with opposite dimer has been added to the figure. See Figure 7e.

22. Electron density maps of both B12 and BV should be included in the extended data for all structures.

The electron density maps of B₁₂ and BV have been added in the SI (see supplementary information Figure S5).

23. Details regarding how the B12 and BV were refined should be added into the methods.

Thanks for the suggestion. We have added details of how the B₁₂ and BV were refined to the method section. See page 10, line 325-326.

24. Details regarding how the kinetics assay were performed are needed – how were the concentrations determined/linear range, etc.

Details of how the kinetics assay were performed have been updated in the method section, please see page 12, lines 386-391.

25. A product control should be added into the LC-MS data shown in Extended Data Figure 6a. The intensities should also be added in this figure.

A standard curve for the product c-di-GMP has been added to updated figure 9. The intensities have been added in the figure.

26. Effort should be made to determine whether the changes are significantly different between samples illustrated in Extended data Figure 6b.

A one-way ANOVA with Tukey test analysis has been run for the data (please see updated figure 9).

27. There should be details regarding the expression of the full length protein and efforts made to characterize its oligomeric state should be included. In addition, details should be included regarding whether the quaternary architecture is impacted by the conditions of the enzymatic assay.

Thanks for the suggestion. We have now included native MS and analytical gel filtration results for the full length *AbDPcob* in apo, dark, and light states. No obvious oligomeric state changes were observed (please see SI figure S1 and S2). We have also included analytical gel filtration data in the presence of GTP to show the quaternary structure remains unchanged during the course of enzymatic turnover.

28. The significant figures for the kinetic data should be consistent.

Thanks for the advice. Significant figures have been modified to keep the consistency (please see updated figure 8).

29. Based on the noted changes for MeCbl in the dark and light shown in Extended data Figure 6b, I'm confused by the authors statement (page 6) regarding "changes mediated through AdoCbl photochemistry are important for this activation process".

When MeCbl binds to the protein, the dark form retains a similar activity to the *apo* protein. However, with AdoCbl bound there is clearly dark inhibition and activation upon exposure to light. Hence, it seems as though the presence of the Ado group and changes mediated through AdoCbl photochemistry (i.e. removal of the Ado group) are important for this inhibition and activation process. We have changed the main text to make it more clear (please see page 6, line 195-197).

30. Page 6 – AlphaFold2 should have a reference.

A reference has now been added (ref 24).

31. Some effort should be made to determine whether the changes are significantly different between samples illustrated in Figure 4b.

A one-way ANOVA with Tukey test analysis has been run for the data (please see updated figure 8).

32. It should be clearly explained whether the purification of these proteins and their incubation with B12 was performed in light or dark conditions.

Details have now been updated in the method section (please see page 9, lines 298 and page 10, lines 311-315).

33. A reference(s) should be added into the methods regarding the source of the extinction coefficients.

These references have now been added (please see page 11, line 346-347, references 32-34).

34. More explanation regarding the use of two different binding equations should be added into the methods.

The reason why two equations have been used was added in the method section (please see page 11, lines 351-353).

35. Figure 4b appears to have the same data for AbDGC in the upper and lower panels but the data looks different with respect to the independent data points.

They are from different measurements. The reason reviewer feels they are same may be that when mixing *AbDGC* and *AbPcob* at 1:1 ratio the light state activity always increases similar ratio when compare with dark state. The raw data have been included as SI table S4-S5.

36. Figure 4b Should show additional data with higher concentrations of *AbPcob* until the activity is saturated.

This is now updated figure 8b. Unfortunately, this is not possible as the DGC activity measurements are performed using an assay that measures the absorbance change at 360 nm. Currently, the highest *AbPcob* concentration we use is 100mM, which is limited by the absorbance of the B12 cofactor at this concentration. Higher concentrations mask the absorbance change at 360 nm and therefore, give rise to unreliable data.

37. There is no c label in Figure 4.

Panel c has been added to this figure, which is now figure 8c.

38. The arrows in panel c of Figure 4 need to be described because not all colors are described.

We have added a description for arrows and all the colours. Please see updated figure legends, Figure 9, page 21, lines 765-766.

39. From the structures, the authors should make and test hypotheses regarding how the transition between dark and light states affects the activity of the DGC domain.

So far, the SAXS data show that conformational changes are important for the activation of DGC activity in *AbDPcob*. This appears to lead to a more flexible DGC domain, which may increase activity (please see page 6-7, lines 202-210). However, unfortunately the lack of a full-length structure makes it difficult to make any hypotheses that we can test. A full-length *AbDPcob* structure would clearly help in this regard and is something we continue to pursue.

40. The illuminations assays and their analysis as does not fully support pure optical coupling (as a thermal component cannot be ruled out). Therefore, I would suggest adding temperature dependence (including lower temperatures) to the LED illumination assays to show that there is something providing energy to get from the red to the green photon. If it's not temperature dependent, the authors should provide some indication of where the energy is coming from.

This is an excellent point and we don't fully understand the mechanism of 'optical coupling' at this stage. As pointed out in our response to reviewer 2, the mechanism of coupling between the 2 cofactors is currently not clear and indeed, may well occur via a different route (e.g. triplet

energy transfer, electron transfer, etc). It will require a lot of further investigation using a range of biophysical spectroscopy and computational chemistry techniques that is beyond the scope of this initial characterisation paper.

41. The analysis conducted in Figures 2a/2b would benefit from SVD-based rank analysis to determine the number of contributing states (see reference 6).

The data shown in figure 2 are static ‘end-point’ absorbance spectra (or difference spectra). The data are included to show the spectral changes observed upon illumination of the dark state with different colours of light. There is no time-resolved data included in these figures and hence, SVD analysis would not be applicable in this instance. We are simply just looking at the ratio of 2 different states (dark and light) with no intermediates.

42. The text (and Table 1) needs to be carefully edited for typos/extra spaces/missing spaces, etc. Some specific examples are given below.

- a. Table 1 is missing (before the Rmerge of the highest resolution shell
- b. Figure 2 is missing spaces before panels c and d
- c. 1.7, 2.0 and 2.3 should be followed by A resolution
- d. His-trap should be HisTrap (made by Cytiva)
- e. Ab is not italicized in figure 4c legend.
- f. Figure 4c legend light blue should be fixed
- g. Page 5lines 162 and 164 appear to refer to Figure 4a not 3

We apologise for the typos and mistakes and they have been corrected in the text now.

43. The figures should be made to match the text such that the Ab or Sas are consistently italicized.

All *Ab* and *Sas* has been italicised in the figures.

Reviewer 4:

Zhang et al present a very interesting study in which they characterize a novel sub-family of the B12-dependent photoreceptors related to the canonical B12 photoreceptor CarH of *Myxococcus xanthus*. In contrast to CarH, which binds adenosylcobalamine (AdoCbl) and acts as transcriptional regulator controlling the expression of genes involved in carotenoid biosynthesis, the characterized, novel photoreceptor sub-family contain an additional globin-like domain as well as different (enzymatic active) ‘effector’ domains such as diguanylate cyclases (DGC). Interestingly, the proteins of the new subfamily (shown for two examples; see below) bind both AdoCbl and the linear tetrapyrrole biliverdin (BV). Given these properties, the authors coined this new subfamily of B12 photoreceptors photocobin (photoactive-cobalamine-bilin, Pcob) proteins. While the authors are the first to experimentally characterize this family of proteins, the family was already identified recently in a bioinformatic study (cited by authors). In detail, the authors studied two proteins of the new subfamily, a ‘standalone’ photoreceptor lacking a fused ‘effector’ domain SasPcob of *Saccharothrix syringae* and a more complex multi-domain protein from an Acidimicrobiaceae bacterium containing a DGC domain as ‘effector’ domain (AbDPcob). The authors verified AdoCbl and BV binding in both proteins, showed that both proteins display light-dependent spectral changes similar to CarH, pointing towards a similar photochemistry. Importantly, when both AdoCbl and BV are bound to the protein, illumination with both red (BV absorption) and green (AdoCbl absorption) light,

trigger photochemical changes in the B12 cofactor. In proteins loaded only with BV and illuminated with red-light, no spectral changes are observed. It thus seems that BV acts as a kind of ‘antenna’ to extend the absorbance range of photocobilins beyond the blue/green region of the spectrum. In addition to detailed spectroscopic studies, the authors also obtained crystal structures of the dark and light state of SasPcob and the AdDPcob photoreceptor domain (without DGC), which provided insight into AdoCbl and BV binding. Since the authors could not obtain a crystal structure of the full-length AbDPcob protein, SAXS was used to study the solution structure of the dark and light state of the protein, revealing significant structural differences. Using the full-length AdDPcob protein, light-dependent DGC activity was verified for the full-length protein. Interestingly, when the isolated DGC domain of AdDPcob was mixed with the isolated photocobalim photoreceptor domains (AdPcob and SasPcob) a light dependent change in DGC activity, which was concentration dependent, was observed, suggesting that in the absence of a fused effector domain standalone photocobalim could control effector activity via protein-protein interactions. Last but not least, bioinformatic studies (e.g. genome proximity analyses) were used to elucidate the potential functions of photocobalim photoreceptors.

The presented work is novel, original and of general interest for the broader biology/microbiology/biophysics community. The work is carried out well, the paper is generally well written and the conclusions are sound. I thus believe the manuscript deserves publication. However, I have a few (rather minor) comments that the authors should consider to improve broader readability beyond the more specialized audience of photobiologists and to place their conclusions on more solid ground.

We would like to thank the referee for their positive comments on our work and are pleased they feel our work is of interest to the broader community. We have now revised the manuscript accordingly as suggested by the reviewer to improve readability further.

Major comments;

I found reading the manuscript rather difficult due to the differentiation between supporting information and extended data. Is there a reason for not placing the extended data into a single supporting information file?

We agree with the reviewer and have now moved all extended data figures in the main text or in the SI to fit the style of the *Nature Comms* journal.

A brief introduction in CarH photochemistry would be needed for readers that are not from the field, i.e. to understand the conclusions drawn by the authors (see page 3, line 98. Discussion about ‘His-on ligation’)

We have included extra text in the introduction to describe CarH photochemistry (see comments above) and have also included more specific text to describe the His-on ligation (see page 4, line 114-118). Figure 7 has also been modified to include panels a-c to compare B₁₂ binding and illustrates the His-on ligation in CarH and PcobS more clearly.

Given the fact, that nothing is known about the natural functions of the photocobalim photoreceptors, can the authors be sure that BV is the natural second chromophore of these proteins? Did the authors test binding of PCB, PEB etc?

We tested PCB also (data not shown), which was also able to bind and is perhaps not surprising due to the similarity in structure. However, heme is not able to bind. As PCB is mainly only found in cyanobacteria and is the main component of phycobiliproteins, this chromophore would not be expected to be found in most of the organisms that contain Pcobs (mainly bacteria). As BV is the bilin type most commonly found in these organisms we focussed on this cofactor to explore the properties of these proteins.

Along those lines, is anything known about bilin synthesis in the host microbes *Saccharothrix syringae* and *Acidimicrobiaceae* bacterium?

Saccharothrix syringae is a mesophilic bacterium and most likely from soil samples (<https://bacdive.dsmz.de/strain>). It has heme and cobalamin biosynthesis modules (<https://www.genome.jp/kegg/>). There is no evidence showing it can synthesis bilin directly, but heme oxygenase (KEGG entry: EKG83_01785, mycobilin-producing) can be found in the genome. For *Acidimicrobiaceae bacterium*, it is an undescribed species in the *Acidimicrobiaceae* family and there is no complete Genome Assembly and Annotation report yet. We couldn't find any of its bilin related metabolite pathway information.

The authors experimentally show that excitation of the BV and AdoCbl loaded proteins at a wavelength were only BV absorbs results in spectral changes resulting from photochemistry of the B12 chromophore, which suggest that BV can act as a type of 'antenna' pigment somewhat reminiscent of the MTHF / FAD system in CPD photolyases and Cry-DASH cryptochromes. However, in those latter systems antenna function is explained by FRET from MTHF to FAD. I don't see how 'antenna' function might be realized in the BV/AdoCbl system. Can the authors speculate how excitation of BV can trigger a CarH like AdoCbl photochemistry?

This is an excellent point and one that will require a lot of further investigation using a range of biophysical spectroscopy and computational chemistry techniques. As the reviewer correctly points out, one would imagine that excitation of BV with red light would mean that singlet state energy transfer to the B12 is unfavourable. Hence, the mechanism of coupling between the 2 cofactors is currently not clear and indeed, may occur via a different route (e.g. triplet energy transfer, electron transfer, etc). Due to this uncertainty, we have been careful not to state what the mechanism might be and just claim that they are optically coupled to avoid any possible confusion. However, clearly there is a lot more work needed in this area to understand the mechanism of coupling.

Please clarify how dark and light state structures were obtained? Did the authors crystallize the proteins in the dark and collected data from dark and illuminated crystals or was the light state preformed and crystallized separately. From reading the methods text 'Crystallisation was performed in the dark using the sitting drop vapour diffusion technique ...' it sounds like the first scenario. Here, on page 3-5 the authors state that the Pcob photoreceptor domain is dimeric (page 3, line 95) in the dark and a monomer in the light state (see page 5, line 139) and that illumination triggers monomerization. I have two questions in this regard:

i) How is such an assessment possible when the proteins were crystallized in the dark? Wouldn't a large-scale structural change such as monomerization break the crystal lattice?

ii) Is there further experimental proof for this assessment e.g. from SAXS studies of the isolated Pcob photoreceptor domains or SEC-MALS? If not, some further proof would be needed.

The details for how we obtained those crystals have been added in the method section (see page 10, line 311-315). The dark crystals were prepared under dimmed red light and then incubated in dark. The light state crystals were obtained by fully illuminating the protein samples prior to crystallisation in the dark. Hence, no photoconversion between the two states *in crystallo* has been carried because, as the reviewer correctly points out, this would not be compatible with the monomerisation process. As for experimental proof for the dimer to monomer change, we have included native MS data for *SasPcob* in the dark and light states (figure S1), which indicate a dimer to monomer change.

SAXS data assessment should be improved: The authors recorded SAXS data for the dark and light state of *AbDPcob* and performed MD simulations of the protein in both states. On page 6 line 187/188 the authors state that ‘AlphaFold2 prediction for *AbDPcob* is compatible with the dark conformation in solution. From the Figure presented in Extended Data set 6, it appears that this conclusion is based on the fact that the full-length model can be fitted into the DAMMIF generated ab initio ‘envelope’. To quantify this assessment, the authors could use CRY SOL to ‘back-calculate’ SAXS data from the AlphaFold2 model and fit this data against the experimental data. This can also be done for selected snapshots from the light state MD simulation to see which structure best fits the experimental data. In this way, the authors might get some information about global light-dependent structural changes.

Thanks for the suggestion. We have compared the theoretical curve calculated using *AbDPcob* dimer.pdb with the experimental data. The same analysis was also carried out on selected light state snapshot. Please see updated SI figure S11.

Apart from genome proximity analyses, some words about the distribution of putative photocobalins would be warranted. In which bacteria are they found? Are there any common physiological traits to these organisms?

Thanks for the suggestion. We have carried out the phylogenetic analysis of *Pcob* proteins. The results indicate that *Pcob* proteins are widely distributed across various types of microorganisms, ranging from archaea to bacteria, more details in supplementary results. No obvious common physiological traits have been found among them.

Minor comments

Page 5, line 140: light-triggered departure of the ... is there a better word for departure? Light-triggered release? photolysis?

Thanks for the suggestion. We have changed the word to “release”. See page 5, line 163.

Page 5, line 162 and line 164: isn't this Figure 4a and not 3a? Extended Data Figure 1 and 2 legend: What are *AbBGD* and *SasBG*? = *AbDPcob* and *SasPcob*?

We apologise for the oversight. It has been corrected (now figure 8a). We have also corrected the text in updated figures 3 and 4.

Extended Data Figure 4 legend: Something doesn't fit here? The figure legend does not seem to match what is shown in the figure. Please clarify!

We apologise for this oversight. This figure was merged with several other figures and the figure legends weren't changed accordingly. This has now been corrected in updated figure 5.

REVIEWER COMMENTS

Reviewer #1 (Remarks to the Author):

I am pleased to see that the authors have addressed the reviewers' comments in a satisfactory manner. I found the clarity of the manuscript significantly enhanced, now mirroring the high level of content and the meticulous work invested in this project. I recommend this manuscript for publication.

Reviewer #2 (Remarks to the Author):

In the revised manuscript " Photocobilins integrate B12 and bilin photochemistry for enzyme control" Shaowei Zhang, Harshwardhan Poddar, Laura Jeffreys, Yuqi Yu, Chuanyang Liu, Kaylee Patel, Linus Johannissen, Lingyun Zhu, Matthew Cliff, Cunyu Yan, Giorgio Schiro, Martin Weik, Michiyo Sakuma, Colin Levy, David Leys, Derren Heyes & Nigel S. Scrutton have given satisfactory answers to many of the important points raised in the reviewing.

Unfortunately, a core subject, integration of B12 and bilin photochemistry for enzyme control (see the title), is not characterized in a satisfactory way. This is regrettable, as the biological function of the described 'photocobilins' is not known. The described 'light-sensing' of BV for the photoreaction of AdoCbl bound to SasPcob may well be (most) relevant, but likewise other biological roles may be the consequence of the structural arrangements of a pair of AdoCbl and bilin molecules bound in close proximity. This latter key structural feature is been established experimentally for the case of SasPcob. As the spectrum of BV in the presence of AbDPcob (Figs. 2b and 3f) is remarkably similar to the one of free BV (Supp.Fig.3b) the reported spectroscopic and other evidence for the proposed structural features of the 'less efficient' BV-binding to AbPcob is not convincing.

The photochemical efficiency of a photocobal AdoCbl photoreaction is not discussed in any detail. A(n even) qualitative or semi-quantitative analysis for the critical situations would be important, as concerns the irradiation at 530 and 660 nm of the AdoCbl-BV-SasPcob complex, in comparison with, e.g., the corresponding situation with AdoCbl bound to SasPcob or AdoCbl bound to CarH, and irradiation at 530 nm. Hence, a major possible consequence of the close proximity of BV is unknown (a possible strong lowering of the quantum efficiency of the B12-photoreaction could be a biologically relevant consequence).

Furthermore, the data shown in Suppl. Figs. 9 and 10 and their delineated analysis do not clearly establish 4',5'-anhydro-adenosine as the sole or major adenosine-type product from the irradiation of either SasPcob or AbDPcob. The chemical shift values extracted from ¹H-NMR spectra depicted in Suppl. Fig. 10 for several H-atoms are close to but do not fit the values reported by Jost et al for isolated 4',5'-anhydro-adenosine and a detailed match with the signals in the spectrum of irradiated CarH is not reported (comparison with authentic 4',5'-anhydro-adenosine would be helpful).

Clearly, the significance of this interesting work would be critically increased by inclusion of further key data and of their analyses satisfying the points raised above.

Reviewer #3 (Remarks to the Author):

The authors did a good job addressing most of my concerns. However, I think they should still allude to the unknowns and specifically spell out that at this point "the mechanism of coupling between the 2 cofactors is currently not clear" to facilitate future studies.

In addition, I believe the CarH-based discovery of the 4',5'- anhydroadenosine product requires an additional reference:

Jost M, Simpson JH, Drennan CL. The Transcription Factor CarH Safeguards Use of Adenosylcobalamin as a Light Sensor by Altering the Photolysis Products. *Biochemistry*. 2015

Supplementary Figure 5 needs to list the sigma at which the maps are contoured (and list the type of maps shown, making sure to include difference electron density maps).

Reviewer #4 (Remarks to the Author):

This is the revised version of a manuscript that I have reviewed before. The authors have greatly improved their manuscript and addressed all of my comments satisfactorily. The edits improve general readability and accessibility for a broader audience. I only have a one minor comment remaining that should be addressed before final acceptance.

Newly added SAXS data analysis: Please add a description to the Materials and Methods section or the Supporting Figure S11 legend how the calculated/theoretical SAXS curves shown in Figure S11 were obtained and provide χ^2 for the fit of the theoretical scattering curves against the experimental data.

REVIEWER COMMENTS

Reviewer 1:

I am pleased to see that the authors have addressed the reviewers' comments in a satisfactory manner. I found the clarity of the manuscript significantly enhanced, now mirroring the high level of content and the meticulous work invested in this project. I recommend this manuscript for publication.

We thank the reviewer for their positive comments on our work and we are pleased that the reviewer is able to recommend publication.

Reviewer 2:

In the revised manuscript “ Photocobilins integrate B12 and bilin photochemistry for enzyme control“ Shaowei Zhang, Harshwardhan Poddar, Laura Jeffreys, Yuqi Yu, Chuanyang Liu, Kaylee Patel, Linus Johannissen, Lingyun Zhu, Matthew Cliff, Cunyu Yan, Giorgio Schiro, Martin Weik, Michiyo Sakuma, Colin Levy, David Leys, Derren Heyes & Nigel S. Scrutton have given satisfactory answers to many of the important points raised in the reviewing.

We are pleased that the reviewer feels we have satisfied many of the original concerns. We have addressed the remaining points below.

Unfortunately, a core subject, integration of B12 and bilin photochemistry for enzyme control (see the title), is not characterized in a satisfactory way. This is regrettable, as the biological function of the described ‘photocobilins’ is not known. The described ‘light-sensing’ of BV for the photoreaction of AdoCbl bound to SasPcob may well be (most) relevant, but likewise other biological roles may be the consequence of the structural arrangements of a pair of AdoCbl and bilin molecules bound in close proximity. This latter key structural feature is been established experimentally for the case of SasPcob. As the spectrum of BV in the presence of AbDPcob (Figs. 2b and 3f) is remarkably similar to the one of free BV (Supp.Fig.3b) the reported spectroscopic and other evidence for the proposed structural features of the ‘less efficient’ BV-binding to AbPcob is not convincing.

We agree with the reviewer that the direct evidence for the binding of BV to AbPcob is less convincing than for SasPcob, and we have mentioned this in the manuscript. It is likely that the spectral change observed upon BV binding in SasPcob is caused by the interaction with Asp270. This residue is absent in AbPcob which is likely the cause of weaker binding and/or less distinctive spectral signature on binding. As stated on page 3, line 83 it is entirely possible that there is a different role for the globin-like domain in these more complex fusion proteins. However, it is clear from Supplementary Figure 4b that when BV is present, illumination with red light elicits a spectral change consistent with BV binding and conversion to the AbPcob light state.

The photochemical efficiency of a photocobalamin AdoCbl photoreaction is not discussed in any detail. A(n even) qualitative or semi-quantitative analysis for the critical situations would be important, as concerns the irradiation at 530 and 660 nm of the AdoCbl-BV-SasPcob complex, in comparison with, e.g., the corresponding situation with AdoCbl bound to SasPcob or AdoCbl

bound to CarH, and irradiation at 530 nm. Hence, a major possible consequence of the close proximity of BV is unknown (a possible strong lowering of the quantum efficiency of the B12-photoreaction could be a biologically relevant consequence).

This is a good point but is difficult to measure experimentally. Although the BV cofactor predominantly absorbs red light, it also has significant absorbance across the entire visible spectrum. Hence, when samples with both BV and B₁₂ bound to the protein are illuminated with green light a mixture of direct B₁₂ excitation and the BV-coupled excitation route is observed. In addition, the presence of both chromophores increases the absorbance at 530nm, meaning that controlled measurements under identical light intensities become even more complicated. That said, we have now conducted additional experiments where we have illuminated single and double chromophore-bound SasPcob with identical light intensities of 530 nm and 660 nm to disentangle the 2 processes. These are shown in supplementary Figure S5 and clearly show that the direct B₁₂ excitation route is more efficient. The presence of both chromophores does appear to slightly reduce the efficiency of the direct B₁₂ excitation route but care must be taken in interpretation of these data due to the points raised above. These findings have been added to the main manuscript (see page 3, lines 97-98)

Furthermore, the data shown in Suppl. Figs. 9 and 10 and their delineated analysis do not clearly establish 4',5'-anhydro-adenosine as the sole or major adenosine-type product from the irradiation of either SasPCob or AbDPcob. The chemical shift values extracted from 1H-NMR spectra depicted in Suppl. Fig. 10 for several H-atoms are close to but do not fit the values reported by Jost et al for isolated 4',5'-anhydro-adenosine and a detailed match with the signals in the spectrum of irradiated CarH is not reported (comparison with authentic 4',5'-anhydro-adenosine would be helpful).

As suggested by the reviewer we have carried out further 1-D and 2-D NMR studies to corroborate our previous findings. These data are presented in supplementary Figures S12-21 and clearly demonstrate that the same photoproduct is formed upon irradiation of SasPcob compared to the archetypal CarH system, also consistent with previous published data (see Jost et al, 2015 for CarH).

Clearly, the significance of this interesting work would be critically increased by inclusion of further key data and of their analyses satisfying the points raised above.

We believe that the inclusion of the new data addresses constructively and definitively the points raised by this reviewer. We thank the reviewer for raising the above, which prompted us to further improve the manuscript.

Reviewer 3:

The authors did a good job addressing most of my concerns. However, I think they should still allude to the unknowns and specifically spell out that at this point "the mechanism of coupling between the 2 cofactors is currently not clear" to facilitate future studies.

We thank the reviewer for their kind words. We have now included this specific statement on page 3, line 101-103.

In addition, I believe the CarH-based discovery of the 4',5'- anhydroadenosine product requires an additional reference:

Jost M, Simpson JH, Drennan CL. The Transcription Factor CarH Safeguards Use of Adenosylcobalamin as a Light Sensor by Altering the Photolysis Products. *Biochemistry*. 2015

We have included this reference in the revised version as reference number 5.

Supplementary Figure 5 needs to list the sigma at which the maps are contoured (and list the type of maps shown, making sure to include difference electron density maps).

2Fo–Fc maps at contour level 1σ were shown in the figure (now Figure S6). The figure legend has been updated to address the point.

Reviewer 4:

This is the revised version of a manuscript that I have reviewed before. The authors have greatly improved their manuscript and addressed all of my comments satisfactorily. The edits improve general readability and accessibility for a broader audience. I only have a one minor comment remaining that should be addressed before final acceptance.

We thank the reviewer for their positive comments on the revised version and we are pleased that the reviewer feels it has improved general readability and accessibility.

Newly added SAXS data analysis: Please add a description to the Materials and Methods section or the Supporting Figure S11 legend how the calculated/theoretical SAXS curves shown in Figure S11 were obtained and provide χ^2 for the fit of the theoretical scattering curves against the experimental data.

This has now been added to the figure legend of Figure S11 (now Figure S23). The theoretical curves were calculated using the molecular dynamics simulation based server WAXSIS (<https://waxsis.uni-saarland.de/> [waxsis.uni-saarland.de]) using the pdb files produced by AlphaFold2. The curves were calculated independently of the experimental patterns, so no fitting procedure is involved against experimental data.

REVIEWERS' COMMENTS

Reviewer #2 (Remarks to the Author):

I am pleased to note the authors have addressed the still existing concerns and have included additional data in their revised manuscript, which should now be recommended for publication.

Interestingly, the now submitted additional Supplementary Figure 5 appears to indicate a noteworthy effect of BV bound to the 'stand alone photoreceptor' SasPcob, by lowering the deduced relative efficiency of the photodecomposition of the bound AdoCbl upon illumination with red light. However, since better quantitative values for the particular chromophore of the bound AdoCbl were not derived from the presented data, the issue of the photochemical efficiency of the holo-SasPcob photoreaction may remain a subject of further experimental analysis and discussion.